# DeepFRC: An End-to-End Deep Learning Model for Functional Registration and Classification

**Siyuan Jiang**[†] **, Yihan Hu**[†] **, Wenjie Li, Pengcheng Zeng**[∗]
Institute of Mathematical Sciences
ShanghaiTech University
Shanghai, China
`{jiangsy2023,huyh1,liwj2022,zengpch}@shanghaitech.edu.cn`

## Abstract

Functional data, representing curves or trajectories, are ubiquitous in fields like biomedicine and motion analysis. A fundamental challenge is *phase variability*—temporal misalignments that obscure underlying patterns and degrade model performance. Current methods often address registration (alignment) and classification as separate, sequential tasks. This paper introduces **DeepFRC**, an end-to-end deep learning framework that jointly learns diffeomorphic warping functions and a classifier within a unified architecture. DeepFRC combines a neural deformation operator for elastic alignment, a spectral representation using Fourier basis for smooth functional embedding, and a class-aware contrastive loss that promotes both intra-class coherence and inter-class separation. We provide the first theoretical guarantees for such a joint model, proving its ability to approximate optimal warpings and establishing a data-dependent generalization bound that formally links registration fidelity to classification performance. Extensive experiments on synthetic and real-world datasets demonstrate that DeepFRC consistently outperforms state-of-the-art methods in both alignment quality and classification accuracy, while ablation studies validate the synergy of its components. DeepFRC also shows notable robustness to noise, missing data, and varying dataset scales. Code is available at `https://github.com/Drivergo-93589/DeepFRC`.

## 1 Introduction

Functional data analysis (FDA) is a key area for analyzing data that varies continuously over domains like time, space, or other variables (Ramsay & Silverman, 2005; Ferraty & Vieu, 2006; Srivastava & Klassen, 2016). Functional data is ubiquitous in fields such as biomechanics, neuroscience, healthcare, and environmental science, appearing in datasets like growth curves, wearable device signals, EEGs, fMRI scans, and air pollution levels. Despite its wide applicability, FDA faces challenges stemming from the infinite-dimensional nature of functional data, as well as issues related to smoothness and misalignment, necessitating advanced analytical tools (Wang et al., 2016; Matuk et al., 2022).

### 1.1 Motivation

Two key tasks in FDA are functional registration (curve alignment) and classification. Registration aligns curves to remove phase variability, enabling meaningful comparisons (Srivastava et al., 2011), while classification assigns labels based on underlying curve features. Traditionally, these tasks are addressed separately, with pre-registration followed by classification. However, this decoupled approach is inefficient, as the label significantly influences the progression pace of the curves, and alignment can provide valuable insights for classification (Liu & Yang, 2009; Tang et al., 2022). Jointly analyzing curve alignment and classification accounts for both temporal and structural variations, offering a more comprehensive understanding of the data.

---

[†] These authors contributed equally.
[∗] Corresponding author: `zengpch@shanghaitech.edu.cn`

Deep learning has revolutionized data analysis across various domains by enabling automatic feature extraction, representation learning, and scalability (LeCun et al., 2015). In FDA, deep learning presents an opportunity to integrate registration and classification into a unified framework. However, its application remains underexplored, with existing studies primarily focused on improving either registration accuracy (Chen & Srivastava, 2021) or classification performance (Yao et al., 2021; Wang & Cao, 2024), but rarely addressing both simultaneously. To address this gap, we propose an end-to-end deep learning framework that combines functional data registration and classification, eliminating the need for separate preprocessing while leveraging neural networks to model the complex, non-linear relationships inherent in functional data.

## 1.2 RELATED WORK

Functional registration is a critical step in FDA, aimed at aligning functional data to correct for phase variability (Ramsay & Silverman, 2005). Traditional approaches include landmark-based methods (Kneip & Gasser, 1992; Ramsay & Silverman, 2005), metric-based methods (Wang & Gasser, 1997; Ramsay & Li, 1998; Srivastava et al., 2011; Srivastava & Klassen, 2016), and model-based methods (Tang & Müller, 2008; Claeskens et al., 2010; Lu et al., 2017). While effective in some contexts, these methods face limitations such as high computational cost, manual intervention, and sensitivity to noise. Landmark-based methods, for example, require subjective and potentially impractical landmark selection (Marron et al., 2015). Recently, neural network-based methods (Chen & Srivastava, 2021) have been proposed, but they are still typically used as preprocessing steps, disconnected from the downstream tasks.

Functional classification has traditionally relied on statistical methods like generalized functional regression (James, 2002; Müller, 2005) and functional principal component analysis (fPCA) (Hall et al., 2000; Leng & Müller, 2006), which involve dimension reduction followed by classification. However, these methods are limited by their reliance on handcrafted features and basis function selection, which restricts their adaptability to complex, non-linear data. The integration of deep learning into functional classification, as seen in works by Thind et al. (2020), Yao et al. (2021), and Wang & Cao (2024), addresses these issues, although they typically treat registration as a separate preprocessing task.

Few studies have integrated registration and classification within a unified framework for FDA. Lohit et al. (2019) proposed a Temporal Transformer Network (TTN) for time series classification, but it is not designed for functional data, neglecting the curve's smoothness and infinite-dimensional nature, thus struggling with accurate alignment. More recently, Tang et al. (2022) introduced a two-level model for joint registration and classification, which employs a parametric mixed-effects model for warping functions (e.g., using B-spline bases with Gaussian assumptions) and requires alternating optimization. However, this approach is constrained by its parametric and distributional assumptions, is computationally expensive, and is not easily extendable to multi-class settings.

## 1.3 CONTRIBUTIONS

Our main contributions are: **(1) Unified Architecture for Joint Learning.** We introduce the first end-to-end deep learning model that integrates a neural deformation operator for diffeomorphic warping, a spectral representation for smooth functional encoding, and a class-aware contrastive loss, enabling mutual reinforcement between alignment and prediction. **(2) Theoretical Foundations.** We establish the first theoretical guarantees for such a model, proving approximation capabilities and providing a generalization bound linking registration fidelity to classification performance. **(3) Extensive Empirical Validation.** Extensive experiments on synthetic and real-world data demonstrate that DeepFRC consistently outperforms state-of-the-art methods in both tasks, with ablation studies confirming the synergy of its components and further analyses highlighting its robustness and computational efficiency.

## 2 THE MODEL

We study supervised learning with functional data, where each sample consists of an observed trajectory and a categorical label. Formally, the dataset is $\{(x_i(\boldsymbol{t}_i), y_i)\}_{i=1}^N$, where $x_i(\boldsymbol{t}_i) = (x_i(t_{i1}), \ldots, x_i(t_{in}))^\intercal$ are observations of an underlying trajectory $x(t)$ sampled at irregular and

potentially misaligned time points $\boldsymbol{t}_i = (t_{i1}, \ldots, t_{in})$, and $y_i \in \{1, \ldots, C\}$ is its class label. We view functional data as trajectories embedded in an infinite-dimensional space, where misalignment corresponds to latent time reparameterizations. Instead of treating alignment and prediction as two separate stages, we propose to jointly learn a diffeomorphic time-warping map $\gamma : t \mapsto \tilde{t}$ and a classifier $f : x(\tilde{t}) \mapsto y$ in a single end-to-end architecture (Figure 1). This allows us to directly learn representations that are invariant to temporal misalignment while remaining discriminative for classification.

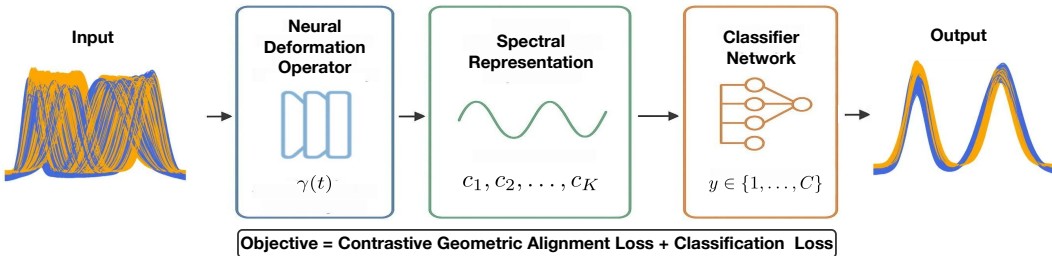

Figure 1: Overview of DeepFRC. Multiple raw functional trajectories are first aligned via a neural deformation operator (1D CNN) that learns diffeomorphic time warping $\gamma(t)$, producing both warped curves and alignment functions. The aligned signals are then expanded in a Fourier basis to obtain spectral coefficients $c_1, \ldots, c_K$, which serve as inputs to a classifier network (MLP with Softmax output) for predicting class labels $y \in 1, \ldots, C$. Training is performed jointly by minimizing a contrastive geometric alignment loss and a classification loss.

## 2.1 NEURAL DEFORMATION OPERATOR FOR TIME WARPING

We introduce a *neural deformation operator*, parameterized by a 1D convolutional network (Kiranyaz et al., 2016; Ince et al., 2016; Kiranyaz et al., 2021), that learns velocity fields defining diffeomorphic warpings. This perspective differs from handcrafted registration methods: instead of explicitly constructing alignment rules, we parameterize $\gamma$ through a deep network optimized jointly with downstream prediction.

Given an input sequence $x_i(\boldsymbol{t}_i) \in \mathbb{R}^n$, a multi-layer 1D CNN extracts temporal features $\tau(x_i(\boldsymbol{t}_i)) \in \mathbb{R}^n$:

$$\boldsymbol{h}^{(l)} = \text{ReLU}(W^{(l)} \circledast \boldsymbol{h}^{(l-1)} + B^{(l)}), \quad l = 1, \ldots, l_1 - 1, \tag{1}$$

with $\boldsymbol{h}^{(0)} = x_i(\boldsymbol{t}_i)$. The final layer is fully connected: $\tau(x_i(\boldsymbol{t}_i)) = \text{ReLU}(W^{(l_1)}\boldsymbol{h}^{(l_1-1)} + \boldsymbol{b}^{(l_1)})$. Following Chen & Srivastava (2021), we use $l_1 = 4$ convolutional blocks with kernel size 3 and channels $16 \to 32 \to 64$. This encoder, parameterized by $\Theta_1$, produces latent features that encode temporal deformation fields.

To construct a boundary-preserving diffeomorphism, we transform the latent features into a monotone cumulative sum: $\tilde{\gamma}_i(t_{ij}) = \frac{\sum_{\mu=0}^{j} \tau_{i\mu}^2}{\sum_{\nu=0}^{n} \tau_{i\nu}^2}$, $\quad j = 1, \ldots, n$, where $t_{i0} = 0, t_{in} = 1$, and $\tau_{i0} = 0$. This ensures $\tilde{\gamma}_i(0) = 0, \tilde{\gamma}_i(1) = 1$, and $\frac{d\tilde{\gamma}_i}{dt} > 0$, satisfying diffeomorphic constraints (Chen & Srivastava, 2021). To further improve smoothness and geometric consistency, we apply an additional normalization:

$$\gamma_i(t_{ij}) = \frac{\sum_{\mu=0}^{j} \tilde{\gamma}_{i\mu}}{\sum_{\nu=0}^{n} \tilde{\gamma}_{i\nu}}. \tag{2}$$

This defines a neural operator that outputs valid, smooth, and data-adaptive warping functions. The construction of $\gamma_i(t)$ via the monotone cumulative sum guarantees a valid diffeomorphism. Squaring the latent features $\tau_i$ ensures non-negative increments, which, after two steps of normalization, enforces strict monotonicity ($\dot{\gamma}_i > 0$) and the boundary conditions $\gamma_i(0) = 0$, $\gamma_i(1) = 1$. In the continuous limit, this yields a $C^1$ function with a positive derivative, fulfilling the requirements for a diffeomorphism on $[0, 1]$.

## 2.2 SPECTRAL REPRESENTATION OF ALIGNED FUNCTIONS

Once trajectories are aligned by $\gamma_i$, we interpolate the aligned function $\tilde{x}_i(t)$ on points $\{(\gamma_i(t_{ij}), x_i(t_{ij}))\}_{j=1}^n$ using stable 1D linear interpolation:

$$\tilde{x}_i(t) \triangleq \sum_{r=0}^{n-1} \left[ x(t_{ir}) + \frac{x_i(t_{i(r+1)}) - x_i(t_{ir})}{\gamma_i(t_{i(r+1)}) - \gamma_i(t_{ir})} (t - \gamma_i(t_{ir})) \right] \cdot \mathbb{1}_{\{\gamma_i(t_{ir}) \leq t \leq \gamma_i(t_{i(r+1)})\}}. \tag{3}$$

Rather than vectorizing $\tilde{x}_i(t)$ into a high-dimensional grid, which ignores smoothness and leads to inefficiency, we embed aligned functions into a compact spectral basis:

$$\tilde{x}_i(t) \approx \sum_{j=1}^K c_{ij} \phi_j(t), \tag{4}$$

where $\{\phi_j(t)\}_{j=1}^K$ are Fourier basis functions and $c_{ij}$ are coefficients estimated by least squares:

$$\tilde{c}_i = \underset{\{c_{ij}\}}{\arg\min} \int_0^1 \left[ \tilde{x}_i(t) - \sum_{j=1}^K c_{ij} \phi_j(t) \right]^2 dt = G^{-1} d_i, \tag{5}$$

with $G_{ij} = \int_0^1 \phi_i(t)\phi_j(t)\,dt$ and $d_{ij} = \int_0^1 \tilde{x}_i(t)\phi_j(t)\,dt$. This spectral representation, inspired by classical functional data analysis, acts as a form of *neural Fourier features*, yielding compact, smooth, and regularized embeddings of aligned functions. We employ 1D linear interpolation (Eq. (3)) for its numerical stability and computational efficiency, and a Fourier basis expansion (Eq. (4)) as it introduces no additional regularization hyperparameters. Crucially, both methods satisfy the Lipschitz continuity condition required by Theorem 3.3.

## 2.3 CLASSIFIER NETWORK

The spectral coefficients $\tilde{c}_i \in \mathbb{R}^K$ are fed into a fully connected classifier with $l_2 - 1$ hidden layers and ReLU activations:

$$h^{(l)} = \text{ReLU}(W^{(l)} h^{(l-1)} + b^{(l)}), \quad l = l_1 + 1, \dots, l_1 + l_2 - 1, \tag{6}$$

followed by a softmax layer: $\psi(\tilde{c}_i) = \text{Softmax}(W^{(l_1+l_2)} h^{(l_1+l_2-1)} + b^{(l_1+l_2)})$. We use a three-layer MLP ($l_2 = 3$) with hidden units of size 8 and 4, parameterized by $\Theta_2$.

## 2.4 OBJECTIVE: CONTRASTIVE GEOMETRIC ALIGNMENT AND CLASSIFICATION

We optimize the model with parameters $\Theta = \{\Theta_1, \Theta_2\}$ by combining geometric alignment and discriminative classification.

**Class-aware Contrastive Alignment.** To align functional trajectories while preserving class structure, we adopt the square-root velocity function (SRVF) representation $q(t) = \text{sign}(\dot{x}(t))\sqrt{|\dot{x}(t)|}$ (Srivastava & Klassen, 2016). For a warped trajectory $x(\gamma(t))$, the SRVF is $(q \star \gamma)(t) = q(\gamma(t))\sqrt{\dot{\gamma}(t)}$. For sample $i$, the observed SRVF vector is $Q_i(\gamma_i) = ((q_i \star \gamma_i)(t_{i1}), \dots, (q_i \star \gamma_i)(t_{in}))^\mathsf{T}$. We define a contrastive alignment loss:

$$\mathcal{L}_1(\Theta_1) = \sum_{j=1}^C \frac{\sum_{i:y_i=j} \|Q_i(\gamma_i) - \bar{Q}^{(j)}\|}{N^{(j)}} + \alpha \sum_{1 \leq u < v \leq C} \|\bar{Q}^{(u)} - \bar{Q}^{(v)}\|^{-1}, \tag{7}$$

where $\bar{Q}^{(j)}$ is the class-wise SRVF mean. The first term encourages intra-class alignment; the second is a contrastive separation term that increases inter-class margins. Together, this yields a new contrastive–geometric alignment objective.

**Classification Loss.** The classifier is trained using standard cross-entropy:

$$\mathcal{L}_2(\Theta) = -\frac{1}{N} \sum_{i=1}^N \sum_{j=1}^C y_{ij} \log \psi_{ij}. \tag{8}$$

---

**Algorithm 1** Training DeepFRC

---

**Require:** Training data $\left\{(x_i(\boldsymbol{t}_i), y_i)\right\}_{i=1}^N$
 1: **Set Hyperparameters**: Size of basis function $K$, loss-related $\{\alpha, \beta\}$, and training parameters $\boldsymbol{\eta}$
    (epochs $E$, batch size, learning rate, etc.)
 2: **Initialize Parameters** $\Theta = \Theta_{\text{initial}}$
 3: **for** $e = 1$ to $E$ **do**
 4:     **Forward Propagation**:
        (1) Compute $\gamma_i(\boldsymbol{t}_i)$ for each $x_i(\boldsymbol{t}_i)$
        (2) Warp $x_i(\boldsymbol{t}_i)$ to obtain $\tilde{x}_i(t)$, calculate its SRVF $Q_i(\boldsymbol{\gamma}_i)$, and extract coefficients $\tilde{\boldsymbol{c}}_i$
        (3) Pass $\tilde{\boldsymbol{c}}_i$ through the classifier to compute $\psi(\tilde{\boldsymbol{c}}_i)$
        (4) Compute the loss $\mathcal{L}(\Theta)$
 5:     **Backward Propagation**: Update $\Theta$ via AdamW optimizer using $\frac{\partial \mathcal{L}(\Theta)}{\partial \theta}, \theta \in \Theta$
 6: **end for**
 7: **Return** Trained DeepFRC with optimized parameters $\Theta^*$

---

**Joint Objective.**  The full objective integrates both alignment and prediction:

$$\mathcal{L}(\Theta) = \mathcal{L}_1(\Theta_1) + \beta \mathcal{L}_2(\Theta), \tag{9}$$

where $\beta$ balances geometric alignment and classification accuracy.

**Summary.**  Integrating Sections 2.1–2.4, we obtain **DeepFRC** (*Deep Functional Registration and Classification*), an end-to-end framework coupling (i) a neural deformation operator for alignment, (ii) a spectral embedding for smooth functional representation, and (iii) a classifier guided by a contrastive–geometric loss (Figure 1). To the best of our knowledge, DeepFRC is the first model to unify alignment and prediction of functional data through diffeomorphic neural operators and contrastive geometry, offering a new paradigm for learning invariant representations of misaligned functional trajectories. A natural extension of our framework allows it to handle $d$-dimensional functional inputs ($d \geq 2$) by simply considering $x_i(\boldsymbol{t}_i) \in \mathbb{R}^{n \times d}$, under the assumption of a shared warping process across dimensions, without requiring additional structural assumptions.

## 2.5 Optimization, Model Selection and Computational Complexity

The objective in Eq. (9) is optimized via stochastic gradient descent over the model parameters $\Theta$, using the AdamW optimizer (Ilya & Frank, 2019) for efficient and stable convergence, particularly in deep architectures (Diederik & Lei, 2015). Gradients with respect to each $\theta \in \Theta$ are computed via the chain rule, as detailed in Eqs. (A.1)–(A.2) of Appendix A. The complete training procedure is outlined in Algorithm 1.

The number of basis functions $K$ is empirically set to 100 and shown to be robust across tasks (Section 4.2). Hyperparameters $\alpha$ and $\beta$, which balance alignment and classification loss terms, are selected via data-splitting methods following (Wang et al., 2023); further model selection details are provided in Appendix B.

DeepFRC achieves linear time complexity $\mathcal{O}(Nn)$ with respect to sample size $N$ and sequence length $n$. In contrast, traditional alignment techniques such as dynamic time warping (DTW), which rely on dynamic programming, incur quadratic cost $\mathcal{O}(Nn^2 k)$ (with $k < n$), rendering them computationally infeasible for long sequences (Sakoe & Chiba, 1978; Chen & Srivastava, 2021).

## 3 Theoretical Analysis

We now discuss the theoretical properties of the proposed model, demonstrating that it achieves low registration error under elastic functional data analysis (EFDA) (Srivastava & Klassen, 2016) framework and low generalization error under mild regularity conditions. Detailed proofs are provided in Appendices C and D.

The phase variability is modeled using monotone and smooth transformations within the set $\Gamma = \{\gamma : [0,1] \to [0,1] \mid \gamma(0) = 0, \gamma(1) = 1, \dot{\gamma} > 0\}$. Under the framework of elastic functional data analysis

(EFDA) (Srivastava et al., 2011), the optimal warpings are defined as: $\boldsymbol{\gamma}^* = \underset{\{\gamma_i\}}{\text{argmin}} Q_{\text{reg}}(\boldsymbol{\gamma}) \triangleq$

$\underset{\{\gamma_i\}}{\text{argmin}} \sum_{j=1}^{C} \frac{\sum_{i \in \{y_i = j\}} \|Q_i(\boldsymbol{\gamma}_i) - \bar{Q}^{(j)}\|}{N^{(j)}}, \gamma_i \in \Gamma$, where "reg" means "registration". For any other estimated

warping $\hat{\boldsymbol{\gamma}}$, the registration error can be quantified as $\Delta Q_{\text{reg}}(\boldsymbol{\gamma}^*, \hat{\boldsymbol{\gamma}}) \triangleq |Q_{\text{reg}}(\boldsymbol{\gamma}^*) - Q_{\text{reg}}(\hat{\boldsymbol{\gamma}})|$.

**Theorem 3.1** (Low Registration Error). *Assume that: (i) The proposed deep neural network has sufficient capacity, (ii) Each warping function $\gamma(t)$ belongs to the admissible set $\Gamma$ and is continuously differentiable, and (iii) The SRVF of $q(t)$ of each curve $x(t)$ is continuous and bounded.*
*Then, for any $\epsilon > 0$, there exists an estimated $\hat{\boldsymbol{\gamma}}$ produced by the model such that:*

$$\Delta Q_{reg}(\boldsymbol{\gamma}^*, \hat{\boldsymbol{\gamma}}) < \epsilon.$$

*Remark* 3.2. Theorem 3.1 provides a theoretical foundation for our approach, establishing that neural networks can approximate smooth warping functions in SRVF space, thereby justifying the use of a learnable diffeomorphic registration module. While direct empirical verification of the approximation error is challenging due to the inability to obtain globally optimal warpings $\boldsymbol{\gamma}^*$ in real data, we validate the practical observability of this result through controlled simulation studies where the ground-truth warpings are known.

We further show that the proposed model achieves a small generalization error. Let $\hat{f}_\Theta : \boldsymbol{x} \rightarrow y$ represent the mapping of the proposed DeepFRC model with a fixed architecture. Define the population risk and empirical risk as $R(\Theta) = \mathbb{E}[l(\hat{f}_\Theta(\boldsymbol{x}), y)]$ and $R_n(\Theta) = \frac{1}{N} \sum_{i=1}^{N} l(\hat{f}_\Theta(\boldsymbol{x}_i), y_i)$, respectively, where $l$ denotes the individual loss function. The generalization error is given by

$$\Delta R_{\text{gen}}(\hat{\Theta}) = |\mathbb{E}_{S,A}[R(\hat{\Theta}) - R_n(\hat{\Theta})]|,$$

where $\hat{\Theta}$ is the parameter set estimated via a random algorithm $A$ based on a random sample $S$. During training, we assume the AdamW optimizer uses an inverse time decay or a custom decay function to ensure the learning rate $\alpha_t$ is monotonically non-increasing with $\alpha_t \leq c_0/t$, for some constant $c_0 > 0$, and that the algorithm runs for $T_0$ steps. The following result builds upon Theorem 3.8 from Hardt et al. (2016) and Theorem 2 from Yao et al. (2021).

**Theorem 3.3** (Low Generalization Error). *Assume that: (i) There exists $\epsilon_0 > 0$, such that $|\bar{Q}^{(u)} - \bar{Q}^{(v)}| \geq \epsilon_0$ for $1 \leq u < v \leq C$ and $\psi_{ij} \geq \epsilon_0$, and (ii) The weight $\Theta$ is restricted to a compact region.*
*Then, for the weights $\hat{\Theta}$ estimated by the proposed model, there exists a constant $c_0 > 0$, such that*

$$\Delta R_{gen}(\hat{\Theta}) \lesssim \frac{T_0^{1-1/c_0}}{N}.$$

*Remark* 3.4. The assumptions of Theorem 3.3 are empirically satisfied and guide model design. Class separation $|\bar{Q}^{(u)} - \bar{Q}^{(v)}| \geq \epsilon_0$ is supported by distinct SRVF means (Figure 3, Table 1). The probability floor $\psi_{ij} \geq \epsilon_0$ is enforced via additively smoothed softmax, keeping all probabilities above $10^{-4}$. Although non-constructive, the bound informs hyperparameter tuning: $\alpha$ is chosen to maximize separation, while $\beta$ balances classification and alignment.

**Discussion of Assumptions.** The assumptions in Theorems 3.1 and 3.3 (Lipschitz continuity, boundedness, compactness) are standard for theoretical analysis. In our framework, these hold by construction: the 1D-CNN uses continuous activations, and training (e.g., with weight decay) implicitly constrains parameters, ensuring the deformation operator yields a Lipschitz-continuous velocity field. The resulting warping function $\gamma(t)$, constructed via the cumulative sum of squared velocities, therefore lies in the admissible set $\Gamma$ with $\dot{\gamma}(t) > 0$. Violations (e.g., from unbounded weights) would break diffeomorphic guarantees and degrade empirical performance, but our experiments confirm that standard training practices suffice to satisfy these conditions, ensuring both theoretical validity and practical robustness.

## 4 EXPERIMENTS

We present the results of functional data registration and classification using DeepFRC, evaluated on both simulated and real-world datasets. Simulated data provides insights into registration, reconstruction, and classification progression during training. Real-world datasets allow for performance comparisons with state-of-the-art methods.

**Evaluation Metrics.** We evaluate our method using metrics for registration, reconstruction, and classification. Registration quality is assessed by the alignment error ($\Delta Q_{\text{reg}}$) between estimated and true warping functions on simulated data, and by the Adjusted Total Variance (ATV) (Chen & Srivastava, 2021) on both simulated and real data, where lower values indicate better alignment. The fidelity of the recovered smooth process is measured via the correlation between true and estimated basis coefficients (simulated data only). Classification performance is reported using accuracy (ACC) and the macro-averaged $F_1$-score (Sokolova & Lapalme, 2009), with higher values being better. Detailed metric definitions are provided in Appendix E.1.

## 4.1 SIMULATION

**Synthetic Data Generation.** We generate a balanced two-class functional dataset $\{x_i(t), y_i\}_{i=1}^N$, where each function $x_i(t)$ combines amplitude and phase variation. Amplitude is modeled by sums of Gaussian bumps, while phase variation is introduced via nonlinear warping functions $\gamma_i(t)$. Class-specific parameters and full generation details are provided in Appendix E.2.

**Results.** Figure 2 demonstrates the iterative improvement of joint registration and classification on a simulated dataset. Initially, the raw curves of two classes are visually inseparable (Fig. 2(a)). Through optimization, DeepFRC progressively refines the alignment, leading to well-separated, class-specific templates (Fig. 2(a)-(d)). This visual improvement is quantified by a rapidly increasing Pearson correlation $\rho(\boldsymbol{c}^*, \hat{\boldsymbol{c}})$, indicating accurate reconstruction of the true, unwarped functions (Fig. 2(e)). Concurrently, both registration and classification metrics improve steadily as the combined loss converges (Fig. 2(f)-(h)), illustrating the synergy between the two tasks. The model's strong generalization is confirmed on a held-out test set (Figure A1, Appendix E.2).

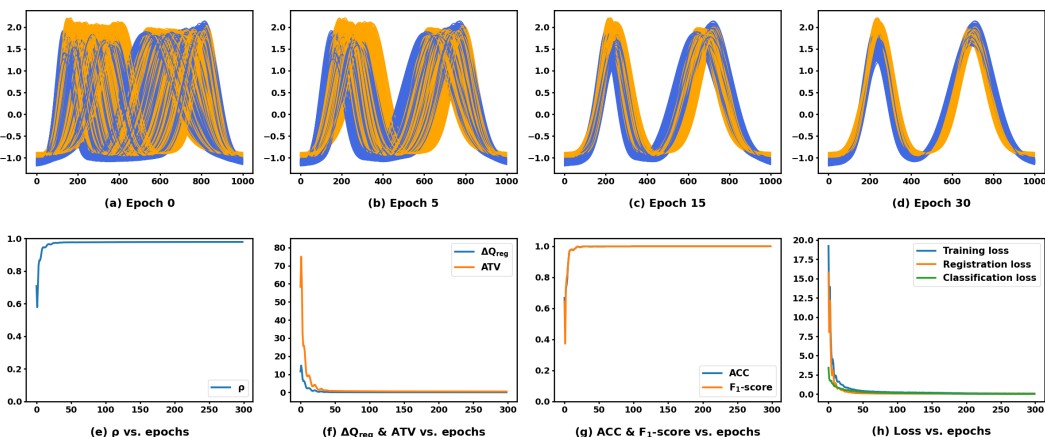

Figure 2: Contribution of iterative optimization in training DeepFRC on simulated two-class (yellow and blue) data. (a)-(d) depict the progression of registration at Epochs 0, 5, 15, and 30. (e)-(h) show the improvement in reconstruction, registration, classification, and loss over SGD epochs.

## 4.2 REAL DATA APPLICATION

**Real-World Datasets.** We evaluate our method on five publicly available datasets, selected for their prevalence in functional data analysis (Ramsay & Silverman, 2002; 2005) and relevance to phase variation: **Wave**, **Yoga**, **Symbol**, and **MotionSense** (Malekzadeh et al., 2019). Wave, Yoga, and Symbol are sourced from the UCR Time Series Classification Archive, while MotionSense is from Kaggle. For the Symbol dataset, we use both binary and three-class subsets. The Wave, Symbol, and MotionSense datasets are class-balanced, whereas Yoga is imbalanced. In terms of dimensionality, Wave, Yoga, and Symbol are one-dimensional, and MotionSense is three-dimensional. Further details are provided in Appendix E.3.

**Baseline Models.** We benchmark DeepFRC against several deep learning models for functional data analysis, including a joint registration-classification model (TTN (Lohit et al., 2019)), a registration-

only model (SrvfRegNet (Chen & Srivastava, 2021)), and five classification models: FCNN$_{raw}$, FCNN$_{fourier}$, FuncNN (Thind et al., 2020), ADAFNN (Yao et al., 2021), and TSLANet (Eldele et al., 2024). To ensure a fair comparison, we evaluate sequential registration-classification pipelines by combining SrvfRegNet with each classification model (excluding TTN, which is already joint). A detailed description of all baseline methods and implementation details are provided in Appendices E.4 and E.5, respectively.

**Performance Comparison.** Table 1 reports both alignment and classification metrics across five real-world datasets. Among all models, only DeepFRC and TTN jointly perform registration and classification. DeepFRC outperforms TTN by using elastic FDA for enhanced functional smoothness, delivering superior registration across all datasets and significant classification improvements on Symbol and MotionSense data. Unlike pipelines combining SrvfRegNet and classification models - which disregard label information during registration - DeepFRC achieves superior alignment, leading to more interpretable transformations and improved downstream classification accuracy. We observe that DeepFRC achieves classification accuracy comparable to TSLANet, a recent state-of-the-art time-series model that outperforms many transformer-based alternatives, across all datasets. This highlights an important trade-off: while powerful classifiers can learn invariance to phase variability from amplitude features alone, DeepFRC's explicit class-aware registration actively enforces this invariance, simplifying the classification task and yielding more interpretable alignments. Figure A2 (Appendix E.5) plots the training loss versus epochs for DeepFRC across all datasets, confirming convergence.

Table 1: Quantitative comparison of registration and classification performance with state-of-the-art approaches across five real datasets. **Bold** indicates best results.

| Model | Wave | | | Yoga | | | Symbol (2 classes) | | | Symbol (3 classes) | | | MotionSense | | |
|---|---|---|---|---|---|---|---|---|---|---|---|---|---|---|---|
| | ATV | ACC | $F_1$-score | ATV | ACC | $F_1$-score | ATV | ACC | $F_1$-score | ATV | ACC | $F_1$-score | ATV | ACC | $F_1$-score |
| DeepFRC | **5.6** | **96.4%** | **0.965** | **16.2** | **89.8%** | **0.909** | **4.8** | **96.0%** | **0.959** | **3.2** | **96.3%** | **0.963** | **25.0** | **95.0%** | **0.952** |
| TTN | 6.3 | 94.7% | 0.948 | 57.7 | 89.4% | 0.904 | 8.6 | 92.0% | 0.918 | 4.5 | 93.3% | 0.933 | 35.1 | 85.0% | 0.857 |
| SrvfRegNet+FCNN$_{raw}$ | 7.3 | 94.6% | 0.947 | 136.0 | 81.0% | 0.830 | 14.8 | 94.5% | 0.942 | 6.5 | 94.7% | 0.947 | 37.7 | 90.0% | 0.909 |
| SrvfRegNet+FCNN$_{fourier}$ | 7.3 | 94.9% | 0.950 | 136.0 | 84.0% | 0.852 | 14.8 | 95.0% | 0.949 | 6.5 | 96.0% | 0.959 | 37.7 | 90.0% | 0.909 |
| SrvfRegNet+FuncNN | 7.3 | 95.7% | 0.957 | 136.0 | 89.0% | 0.908 | 14.8 | 93.5% | 0.933 | 6.5 | 94.7% | 0.947 | 37.7 | 90.0% | 0.889 |
| SrvfRegNet+ADAFNN | 7.3 | 94.6% | 0.949 | 136.0 | 73.4% | 0.753 | 14.8 | 89.0% | 0.894 | 6.5 | 94.3% | 0.943 | 37.7 | 85.0% | 0.857 |
| SrvfRegNet+TSLANet | 7.3 | **96.4%** | 0.961 | 136.0 | 89.3% | 0.884 | 14.8 | 95.5% | 0.955 | 6.5 | **96.3%** | 0.960 | 37.7 | **95.0%** | **0.952** |

**Alignment Visualization and Interpretability.** Figure 3 compares alignment quality on the Symbol (3 classes) dataset. DeepFRC produces smooth, class-separated functional alignments, while TTN distorts class-specific trajectories (e.g., purple), and SrvfRegNet loses inter-class separation (e.g., blue vs. yellow). Both baselines significantly degrade interpretability (see also Figures A3 and A4, Appendix E.5). DeepFRC's superior alignment improves inference reliability and preserves generalization performance - critical for real-world applications. Beyond quantitative metrics, these alignments offer significant practical interpretability. For **MotionSense**, DeepFRC's warps synchronize biomechanical events like heel-strikes and mid-swings, producing crisp, physiologically meaningful activity profiles. For **Symbol**, it explicitly corrects for variable writing speeds and stroke orders, normalizing user-specific kinetics to reveal canonical symbol shapes. This ability to distill noisy, misaligned observations into clean, class-specific functional templates demonstrates the method's utility not only for boosting classification accuracy but also for providing domain experts with intelligible and representative data patterns.

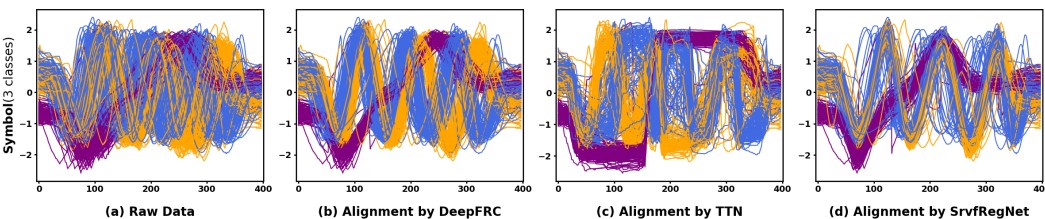

Figure 3: Visual comparison of alignment on the Symbol (3 classes).

**Ablation Studies.** The ablation study in Table 2 evaluates the individual contributions of the core components of our method: the Neural Deformation Operator (N.D.O.) for registration, the

Spectral Representation (S.R.), and the Classifier Network (C.N.). The results demonstrate a clear synergy between these modules. Removing the registration component (DeepFRC w/o N.D.O.) significantly degrades classification performance, particularly on the Yoga, Symbol (2-class), and MotionSense datasets. Conversely, removing the classifier (DeepFRC w/o C.N.) impairs registration accuracy. The removal of the spectral representation (DeepFRC w/o S.R.) adversely affects both tasks across all datasets. These findings underscore the importance of the spectral representation and the tight coupling between registration and classification within DeepFRC's unified architecture. The performance advantages of the full DeepFRC model are statistically significant. As reported in Table 3, paired $t$-tests across 10 random seeds confirmed that removing any core component leads to significant performance degradation ($p < 0.05$ for all relevant metrics across all datasets; $p < 0.01$ for the vast majority of tests). This provides robust statistical evidence that the Neural Deformation Operator, Spectral Representation, and joint optimization are all essential to the framework's success.

Table 2: Ablation study: contributions of three components in DeepFRC.

| Model | Wave | | | Yoga | | | Symbol (2 classes) | | | Symbol (3 classes) | | | MotionSense | | |
|---|---|---|---|---|---|---|---|---|---|---|---|---|---|---|---|
| | ATV | ACC | $F_1$-score | ATV | ACC | $F_1$-score | ATV | ACC | $F_1$-score | ATV | ACC | $F_1$-score | ATV | ACC | $F_1$-score |
| DeepFRC | 5.6 | 96.4% | 0.965 | 16.2 | 89.8% | 0.909 | 4.8 | 96.0% | 0.959 | 3.2 | 96.3% | 0.963 | 25.0 | 95.0% | 0.952 |
| DeepFRC w/o N.D.O. | – | 94.4% | 0.946 | – | 83.1% | 0.846 | – | 91.0% | 0.905 | – | 94.7% | 0.947 | – | 90.0% | 0.909 |
| DeepFRC w/o S.R. | 5.8 | 95.3% | 0.955 | 17.7 | 89.2% | 0.903 | 5.3 | 94.5% | 0.945 | 3.3 | 93.3% | 0.933 | 28.8 | 90.0% | 0.900 |
| DeepFRC w/o C.N. | 7.3 | – | – | 136.0 | – | – | 14.8 | – | – | 6.5 | – | – | 37.7 | – | – |

Table 3: P-values from paired t-tests (10 runs) comparing full DeepFRC against ablated variants. Bold indicates significance ($p < 0.05$).

| Hypothesis Tests | Wave | | | Yoga | | | Symbol (2 classes) | | | Symbol (3 classes) | | | MotionSense | | |
|---|---|---|---|---|---|---|---|---|---|---|---|---|---|---|---|
| ($p$-value) | ATV | ACC | $F_1$-score | ATV | ACC | $F_1$-score | ATV | ACC | $F_1$-score | ATV | ACC | $F_1$-score | ATV | ACC | $F_1$-score |
| Full vs. w/o N.D.O. | – | **0.0045** | **0.0051** | – | **0.0023** | **0.0033** | – | **0.0005** | **0.0006** | – | **0.0000** | **0.0000** | – | **0.0000** | **0.0000** |
| Full vs. w/o S.R. | **0.0453** | **0.0081** | **0.0080** | **0.0370** | **0.0349** | **0.0449** | **0.0382** | **0.0150** | **0.0179** | **0.0056** | **0.0001** | **0.0003** | **0.0025** | **0.0001** | **0.0000** |
| Full vs. w/o C.N. | **0.0000** | – | – | **0.0000** | – | – | **0.0000** | – | – | **0.0000** | – | – | **0.0000** | – | – |

**Sensitivity Analysis.** We assess DeepFRC's robustness to the choice of basis family and size $K$ across five real datasets in Figure 4. The left panel shows performance is stable across basis types (Fourier, B-spline, polynomial). We attribute this robustness to two factors: (a) joint training adapts alignment to the inductive biases of each basis, and (b) class-aware registration loss guides the model toward discriminative structures, reducing basis-specific limitations. The right panel shows that increasing $K$ improves performance up to around $K = 100$, after which it plateaus. We believe this invariance arises from (a) upstream registration reducing functional variability, and (b) downstream layers adaptively re-weighting or pruning redundant basis functions.

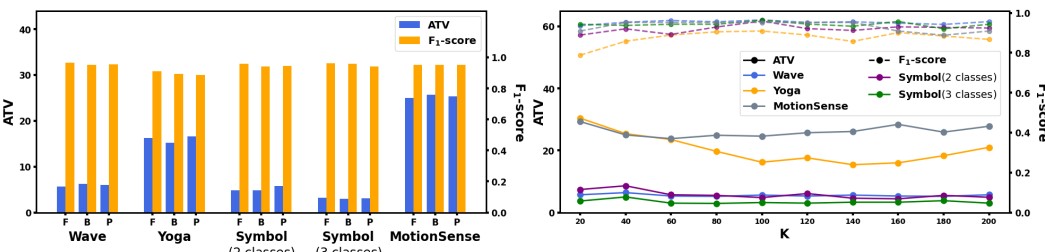

Figure 4: **Left:** Sensitivity to basis family. "F": Fourier; "B": B-spline; "P": Polynomial. **Right:** Sensitivity to number of basis functions $K$.

## 5 DISCUSSION

**Computational Efficiency.** DeepFRC offers significant computational advantages during training, with an average runtime of **60s** on our real-world datasets—dramatically faster than the traditional registration method DTW (**1000s**) and competitive with deep baselines like TTN (**15s**) and SrvfRegNet hybrids (**90–100s**). This efficiency is achieved by eschewing pairwise dynamic programming. Although training time is on par with other deep models, DeepFRC achieves superior accuracy, striking an excellent trade-off between performance and computational cost. A detailed analysis of

training complexity shows DeepFRC requires an average of 1000 TFLOPs with 200k parameters, representing a favorable trade-off between the high accuracy of SrvfRegNet hybrids (1500 TFLOPs, 200k params) and the lower capability of TTN (200 TFLOPs, 100k params).

**Robustness to Noise and Reconstruction Accuracy.** We evaluate robustness by injecting Gaussian noise ($\sigma = 0, 0.05, 0.10, 0.20$) into synthetic data and measuring reconstruction quality via the Pearson correlation between true and estimated basis coefficients. As shown in Table A3, DeepFRC maintains high correlation under low-to-moderate noise, outperforming TTN and exhibiting greater stability than SrvfRegNet. Even under high noise ($\sigma = 0.20$), its performance degrades gracefully while surpassing competitors, indicating that the smooth basis representation and joint optimization confer strong resilience.

**Scalability to Large and Small Data.** DeepFRC scales efficiently to large datasets with $\mathcal{O}(Nn)$ complexity (Section 2.5). On the augmented Symbol dataset ($100 \times$), it achieves near-raw performance (ATV=4.8, $F_1$-score=0.944 vs. 4.8, 0.959) and outperforms all baseline models in both alignment and classification (Table A4). This demonstrates that its performance advantages are maintained at scale. Simulations under sparse data conditions (samples, time points, or both) confirm robust classification, though registration requires sequence length $n \geq 100$ for reliability (Table A5, Appendix E.5).

**Robustness to Missing Data.** We evaluated DeepFRC's tolerance to missing observations by randomly removing 5-10% of data points from half of each real-world dataset (Section 4.2), imputing gaps via Fourier splines (Wahba, 1990). Under identical hyperparameters, DeepFRC achieved comparable registration and classification performance to complete data (Figure A5, Appendix E.5 ).

**Robustness to Non-Diffeomorphic Warpings.** To assess the sensitivity of DeepFRC to violations of the diffeomorphic assumption, we followed the same synthetic data generation protocol as in the main experiments and constructed additional test sets in which a proportion of curves were warped by mappings containing flat regions (i.e., intervals where $\gamma'(t) = 0$) covering a specified fraction of the domain. These perturbations mimic temporal "jumps" and constitute clear departures from strict monotonicity. As reported in Table A6 (Appendix E.5), DeepFRC exhibits graceful degradation under such violations: classification accuracy remains above $90\%$ even when $20\%$ of the curves contain flat regions spanning $60\%$ of their domain. This indicates that the model learns a smooth, diffeomorphism-like approximation to the mis-specified warpings. Although alignment quality (ATV) is naturally more sensitive to these distortions, the discriminative structure captured by the spectral features remains largely preserved.

## 6 CONCLUSION

We introduced **DeepFRC**, a unified framework for joint functional registration and classification with theoretical guarantees and comprehensive empirical validation. DeepFRC consistently outperforms state-of-the-art baselines and remains robust across real-world and simulated settings with noise, missing values, and scale variation. While empirically invariant to the choice of basis functions, its theoretical underpinnings remain open.

Our work also highlights key limitations and considerations for joint modeling. DeepFRC's performance may degrade on highly volatile trajectories with sharp, discontinuous changes, as it relies on smooth, diffeomorphic warping assumptions. Furthermore, the class-aware alignment can be sensitive to significant label noise. A joint framework is not universally optimal; separate processing remains preferable when phase variability is minimal, label information is unreliable, or a powerful classifier already exhibits strong inherent invariance to misalignment.

Future work will focus on adaptive architectures and robust loss designs that explicitly capture signal complexity to enhance generalization, particularly for challenging non-smooth functions and noisy label settings.

ACKNOWLEDGMENTS

This work was supported by the National Natural Science Foundation of China (NSFC) Youth Science Fund [Grant No. 12301353] and the High-Performance Computing (HPC) platform at ShanghaiTech University. We also acknowledge the use of DeepSeek and ChatGPT to refine the language and enhance the clarity of this paper.

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

## A  OBJECTIVE FUNCTION GRADIENT

By the chain rule, the gradient of the objective function with respect to any $\theta \in \Theta$ is derived as follows:

$$\frac{\partial \mathcal{L}(\Theta)}{\partial \theta} = \begin{cases} 2 \sum_{j=1}^{C} \frac{\sum_{i \in \{y_i = j\}} \left( Q_i(\boldsymbol{\gamma}_i) \right)^{\mathsf{T}} \cdot \frac{\partial Q_i(\boldsymbol{\gamma}_i)}{\partial \gamma_i} \cdot \frac{\partial \gamma_i}{\partial \theta}}{N^{(j)}} - \frac{\beta}{N} \sum_{i=1}^{N} \sum_{j=1}^{C} \frac{y_{ij}}{\psi_{ij}} \cdot \frac{\partial \psi_{ij}}{\partial \theta}, & \theta \in \Theta_1, \\ -\frac{\beta}{N} \sum_{i=1}^{N} \sum_{j=1}^{C} \frac{y_{ij}}{\psi_{ij}} \cdot \frac{\partial \psi_{ij}}{\partial \theta}, & \theta \in \Theta_2. \end{cases} \tag{A.1}$$

Here, the gradient of the warped SRVF with respect to the warping function $\gamma_i$ is given by:

$$\frac{\partial Q_i(\gamma_i)}{\partial \gamma_i}[k] = \dot{q}_i(\gamma_i(t_{ik})) \sqrt{\dot{\gamma}(t_{ik})} + q_i(\gamma_i(t_{ik})) \frac{\ddot{\gamma}(t_{ik})}{2\sqrt{\dot{\gamma}(t_{ik})}}, \quad k = 1, \dots, n. \tag{A.2}$$

When computing the gradient of $\mathcal{L}_1(\Theta_1)$ with respect to the warped SRVF ($Q_i(\boldsymbol{\gamma}_i)$), the mean SRVF ($\bar{Q}^{(j)}$) is treated as constant, updated using the arithmetic mean of the warped SRVF from the previous iteration (Chen & Srivastava, 2021). For $\theta \in \Theta_1$, the gradient $\frac{\partial \gamma_i}{\partial \theta}$ can be expressed as the product of gradients from integration, fully connected layers, and the 1D-CNN module. Similarly, $\frac{\partial \psi_{ij}}{\partial \theta}$ is derived from the gradient of $l_2$ fully connected layers, interpolation, integration, and the 1D-CNN module (Yann et al., 2015). For $\theta \in \Theta_2$, $\frac{\partial \psi_{ij}}{\partial \theta}$ involves the gradient from fully connected layers in the prediction module (Yann et al., 2015).

## B  MODEL SELECTION OF DEEPFRC

The model's hyperparameters include the size of basis functions $K$, loss-related parameters $\{\alpha, \beta\}$, and training-related parameters (such as epoch size $E$, batch size, and learning rate, denoted by vector $\boldsymbol{\eta}$). For the basis representation module, we set $K = 100$ with a Fourier basis, and the sensitivity analysis will be studied in real data analysis. To select the other hyperparameters, we follow the procedure proposed by Wang et al. (2023): the dataset $\{(x(\boldsymbol{t}_i), y_i)\}_{i=1}^{N}$ is firstly split into two subsets with a 4:1 ratio. For each combination of hyperparameters, the model is trained by minimizing $\mathcal{L}_{\text{train}}$ (Eq. (9)) on the larger subset, and testing error $\mathcal{L}_{\text{test}}$ is computed on the smaller subset. The combination minimizing $\mathcal{L}_{\text{test}}$ is selected.

In the 1D-CNN module of our model, we fixed the setting of the network structure (set $l_1 = 4$, and set kernel sizes as 3-3-3 and the channel dimensions as 16-32-64 for the three hidden convolutional layers). To improve efficiency, robustness, and interpretability, we apply max-pooling (Nagi et al., 2011) and 1D batch normalization (Ioffe & Szegedy, 2015) after each layer to downsample inputs and stabilize training, and use global averaging (Krizhevsky et al., 2017) followed by the final fully connected layer to further reduce features and noise. In the classifier module, we set $l_2 = 3$ and nodes number $\boldsymbol{n} = (8, 4)$. The above configuration for 1D-CNN module and the fully connected neural network classifier is widely used Chen & Srivastava (2021); Lohit et al. (2019), and works well across all the datasets in this paper.

## C  PROOF OF THEOREM 3.1

For simplicity, we assume $\mathbf{t}_i = \left( \frac{0}{T}, \cdots, \frac{T}{T} \right)$, where $[0, T]$ is the observed time range, in the following proof. Consider the function $f(x) = \sqrt{x}$, which is continuous on $[0, 1]$. For any $\varepsilon > 0$, there exists $\delta_1 > 0$ such that for all $|x_1 - x_2| < \delta_1$, we have $\left| \sqrt{x_1} - \sqrt{x_2} \right| < \varepsilon$. Similarly, since $q$ is continuous, for any $\varepsilon > 0$, there exists $\delta_2 > 0$ such that for all $|x_1 - x_2| < \delta_2$, we have $|q(x_1) - q(x_2)| < \varepsilon$.

Let $\mathscr{G}$ be the domain of the neural network's output function, and $\Gamma$ be the domain of the alignment functions $\gamma$. Denote $\Phi : \mathscr{G} \to \Gamma$ as the mapping from the neural network output to the alignment functions. Since both input and output of the network are discrete, we consider values only at discrete points. We define $g_i^{(k)} = g_i \left( \frac{k}{T} \right)$ to maintain consistency between discrete vectors and continuous functions. The mapping is defined as:

$$\tau_i \mapsto \tilde{\gamma}_i \mapsto \gamma_i,$$

$$\tilde{\gamma}_i\left(\frac{k}{T}\right) = \frac{\sum_{s=0}^k \left(\tau_i^{(s)}\right)^2}{\sum_{s=0}^T \left(\tau_i^{(s)}\right)^2}, \quad \gamma_i\left(\frac{k}{T}\right) = \frac{\sum_{s=0}^k \tilde{\gamma}_i^{(s)}}{\sum_{s=0}^T \tilde{\gamma}_i^{(s)}}.$$

Let $\tau_i$ be such that $\Phi(\tau_i) = \gamma_i$. Since $\Phi(t\tau_i) = \Phi(\tau_i)$ for all $t \neq 0$, we can assume, without loss of generality, that $\|g_i\|_2 = 1$.

By the approximation property of convolutional neural networks (Yarotsky, 2018) and classical universal approximation results (Cybenko, 1989; Funahashi, 1989; Hornik, 1991; Stinchcombe, 1999), for any $\delta_4 > 0$, there exists a neural network $NN_\Theta$ with parameters $\Theta$ such that

$$\sup_i \|NN_\Theta(\mathbf{x}_i) - \tau_i\|_2 < \delta_4.$$

Let $\hat{\tau}_i = NN_\Theta(\mathbf{x}_i)$. We now aim to show that for sufficiently small $|\tau_i - \hat{\tau}_i|$, we also have $|\gamma_i - \hat{\gamma}_i|$ sufficiently small.

$$\left|\tilde{\gamma}_i\left(\frac{k}{T}\right) - \hat{\tilde{\gamma}}_i\left(\frac{k}{T}\right)\right| = \left|\frac{\sum_{s=0}^k \tau_i^2\left(\frac{s}{T}\right)}{\sum_{s=0}^T \tau_i^2\left(\frac{s}{T}\right)} - \frac{\sum_{s=0}^k \hat{\tau}_i^2\left(\frac{s}{T}\right)}{\sum_{s=0}^T \hat{\tau}_i^2\left(\frac{s}{T}\right)}\right|$$

$$\leq \left|\frac{\sum_{s=0}^k \tau_i^2\left(\frac{s}{T}\right)}{\sum_{s=0}^T \tau_i^2\left(\frac{s}{T}\right)} - \frac{\sum_{s=0}^k \hat{\tau}_i^2\left(\frac{s}{T}\right)}{\sum_{s=0}^T \tau_i^2\left(\frac{s}{T}\right)}\right| + \left|\frac{\sum_{s=0}^k \hat{\tau}_i^2\left(\frac{s}{T}\right)}{\sum_{s=0}^T \tau_i^2\left(\frac{s}{T}\right)} - \frac{\sum_{s=0}^k \hat{\tau}_i^2\left(\frac{s}{T}\right)}{\sum_{s=0}^T \hat{\tau}_i^2\left(\frac{s}{T}\right)}\right|$$

$$\leq \sum_{s=0}^k \left(\tau_i\left(\frac{s}{T}\right) + \hat{\tau}_i\left(\frac{s}{T}\right)\right)\delta_4 + (1+\delta_4)^2 \frac{(2+\delta_4)\delta_4}{(1-\delta_4)^2}$$

$$\leq \sum_{s=0}^k \left(\tau_i\left(\frac{s}{T}\right) + \hat{\tau}_i\left(\frac{s}{T}\right)\right)\delta_4 + 27\delta_4 = M_k^{(1)}\delta_4.$$

Therefore, we have

$$\|\tilde{\gamma}_i - \hat{\tilde{\gamma}}_i\|_2 \leq \sqrt{\sum_{k=0}^T M_k^1} = M_{-1}^{(1)}\delta_4.$$

By Hölder's inequality $\|f\|_2 \leq \|f\|_1 \leq \sqrt{T+1}\|f\|_2$, we have

$$\|\tilde{\gamma}_i - \hat{\tilde{\gamma}}_i\|_1 \leq \sqrt{T+1}M_{-1}^{(1)}\delta_4.$$

Note that $\|\tilde{\gamma}_i\|_1 \geq \tilde{\gamma}_i\left(\frac{T}{T}\right) = 1$. On the other hand, for $\gamma_i$, we have

$$\left|\gamma_i\left(\frac{k}{T}\right) - \hat{\gamma}_i\left(\frac{k}{T}\right)\right| = \left|\frac{\sum_{s=0}^k \tilde{\gamma}_i\left(\frac{s}{T}\right)}{\sum_{z=0}^T \tilde{\gamma}_i\left(\frac{z}{T}\right)} - \frac{\sum_{s=0}^k \hat{\tilde{\gamma}}_i\left(\frac{s}{T}\right)}{\sum_{z=0}^T \hat{\tilde{\gamma}}_i\left(\frac{z}{T}\right)}\right|$$

$$= \left|\frac{\sum_{s=0}^k \tilde{\gamma}_i\left(\frac{s}{T}\right)}{\sum_{z=0}^T \tilde{\gamma}_i\left(\frac{z}{T}\right)} - \frac{\sum_{s=0}^k \hat{\tilde{\gamma}}_i\left(\frac{s}{T}\right)}{\sum_{z=0}^T \hat{\tilde{\gamma}}_i\left(\frac{z}{T}\right)} + \frac{\sum_{s=0}^k \hat{\tilde{\gamma}}_i\left(\frac{s}{T}\right)}{\sum_{z=0}^T \hat{\tilde{\gamma}}_i\left(\frac{z}{T}\right)} - \frac{\sum_{s=0}^k \hat{\tilde{\gamma}}_i\left(\frac{s}{T}\right)}{\sum_{z=0}^T \tilde{\gamma}_i\left(\frac{z}{T}\right)}\right|$$

$$\leq \left|\frac{\sum_{s=0}^k \tilde{\gamma}_i\left(\frac{s}{T}\right)}{\sum_{z=0}^T \tilde{\gamma}_i\left(\frac{z}{T}\right)} - \frac{\sum_{s=0}^k \hat{\tilde{\gamma}}_i\left(\frac{s}{T}\right)}{\sum_{z=0}^T \tilde{\gamma}_i\left(\frac{z}{T}\right)}\right| + \left|\frac{\sum_{s=0}^k \hat{\tilde{\gamma}}_i\left(\frac{s}{T}\right)}{\sum_{z=0}^T \tilde{\gamma}_i\left(\frac{z}{T}\right)} - \frac{\sum_{s=0}^k \hat{\tilde{\gamma}}_i\left(\frac{s}{T}\right)}{\sum_{z=0}^T \hat{\tilde{\gamma}}_i\left(\frac{z}{T}\right)}\right|$$

$$\leq \left|\frac{\sum_{s=1}^k \left(\tilde{\gamma}_i\left(\frac{s}{T}\right) - \hat{\tilde{\gamma}}_i\left(\frac{s}{T}\right)\right)}{\sum_{z=0}^T \tilde{\gamma}_i\left(\frac{z}{T}\right)}\right| + \|\hat{\tilde{\gamma}}_i\|_1 \left|\frac{\sum_{s=0}^k \left(\hat{\tilde{\gamma}}_i\left(\frac{s}{T}\right) - \tilde{\gamma}_i\left(\frac{s}{T}\right)\right)}{\left(\sum_{z=0}^T \tilde{\gamma}_i\left(\frac{z}{T}\right)\right)\left(\sum_{z=0}^T \hat{\tilde{\gamma}}_i\left(\frac{z}{T}\right)\right)}\right|$$

$$\leq \frac{k\sqrt{T+1}M_{-1}^{(1)}\delta_4}{\|\tilde{\gamma}_i\|_1} + \left(\|\tilde{\gamma}_i\|_1 + \sqrt{T+1}M_{-1}^{(1)}\delta_4\right)\frac{\delta_4}{\|h_i\|_1\left(\|\tilde{\gamma}_i\|_1 - \sqrt{T+1}M_{-1}^{(1)}\delta_4\right)}$$

$$\leq M_k^{(2)}\delta_4 \quad \text{for some } M_k^{(2)}.$$

Such that we have

$$\|\gamma_i - \hat{\gamma}_i\|_2 \leq \sqrt{\sum_{k=0}^T M_k^{(2)}} := M_{-1}^{(2)}\delta_4.$$

Recall that $Q_i(\gamma_i)\left(\frac{k}{T}\right) = (q_i \circ \gamma_i\left(\frac{k}{T}\right)) \cdot \sqrt{\dot{\gamma}_i\left(\frac{k}{T}\right)}$. Using the conditions that $q_i$ and $\gamma_i$ are bounded, we can bound the difference between $Q_i(\gamma_i)$ and $Q_i(\hat{\gamma}_i)$ by:

$$
\left| Q_i(\gamma_i)\left(\frac{k}{T}\right) - Q_i(\hat{\gamma}_i)\left(\frac{k}{T}\right) \right|
$$

$$
\leq \left| q_i \circ \gamma_i\left(\frac{k}{T}\right) \cdot \left( \sqrt{\dot{\gamma}_i\left(\frac{k}{T}\right)} - \sqrt{\dot{\hat{\gamma}}_i\left(\frac{k}{T}\right)} \right) \right| + \left| \sqrt{\dot{\hat{\gamma}}_i\left(\frac{k}{T}\right)} \cdot \left( q_i \circ \gamma_i\left(\frac{k}{T}\right) - q_i \circ \hat{\gamma}_i\left(\frac{k}{T}\right) \right) \right|
$$

$$
\leq \sup_t |q_i(t)| \cdot \left| \sqrt{\dot{\gamma}_i\left(\frac{k}{T}\right)} - \sqrt{\dot{\hat{\gamma}}_i\left(\frac{k}{T}\right)} \right| + \sup_t \sqrt{\dot{\gamma}_i(t)} \cdot \left| q_i\left(\gamma_i\left(\frac{k}{T}\right)\right) - q_i\left(\hat{\gamma}_i\left(\frac{k}{T}\right)\right) \right|.
$$

We use $\frac{\hat{\gamma}_i\left(\frac{k+1}{T}\right) - \hat{\gamma}_i\left(\frac{k-1}{T}\right)}{\frac{2}{T}}$ to approximate the derivative for $\hat{\gamma}_i$, so we have

$$
\left| \dot{\gamma}_i\left(\frac{k}{T}\right) - \dot{\hat{\gamma}}_i\left(\frac{k}{T}\right) \right| \leq \left| \dot{\gamma}_i\left(\frac{k}{T}\right) - \frac{\gamma_i\left(\frac{k+1}{T}\right) - \gamma_i\left(\frac{k-1}{T}\right)}{\frac{2}{T}} \right| + \left| \frac{\gamma_i\left(\frac{k+1}{T}\right) - \gamma_i\left(\frac{k-1}{T}\right)}{\frac{2}{T}} - \frac{\hat{\gamma}_i\left(\frac{k+1}{T}\right) - \hat{\gamma}_i\left(\frac{k-1}{T}\right)}{\frac{2}{T}} \right|
$$

$$
\leq \left| \dot{\gamma}_i\left(\frac{k}{T}\right) - \frac{\gamma_i\left(\frac{k+1}{T}\right) - \gamma_i\left(\frac{k-1}{T}\right)}{\frac{2}{T}} \right| + T \sup_k \left| \gamma_i\left(\frac{k}{T}\right) - \hat{\gamma}_i\left(\frac{k}{T}\right) \right|.
$$

For $0 < \delta_3 < 1$, there exists a $T > 0$ such that:

$$
\left| \dot{\gamma}_i\left(\frac{k}{T}\right) - \frac{\gamma_i\left(\frac{k+1}{T}\right) - \gamma_i\left(\frac{k-1}{T}\right)}{\frac{2}{T}} \right| < \delta_3, \quad \forall k.
$$

Here we can choose $\delta_3 < \frac{\delta_1}{2}$, $\delta_2 = \min\{\delta_2, \frac{\delta_1}{2T}\}$, and $\delta_4 < \delta_2$. Then, we have

$$
\left| \sqrt{\dot{\gamma}_i\left(\frac{k}{T}\right)} - \sqrt{\dot{\hat{\gamma}}_i\left(\frac{k}{T}\right)} \right| < \varepsilon, \quad \left| q_i \circ \gamma_i\left(\frac{k}{T}\right) - q_i \circ \hat{\gamma}_i\left(\frac{k}{T}\right) \right| < \varepsilon,
$$

such that

$$
\left| Q_i(\gamma_i)\left(\frac{k}{T}\right) - Q_i(\hat{\gamma}_i)\left(\frac{k}{T}\right) \right| \leq M_{-1}^{(3)} \varepsilon, \quad \text{where } M_{-1}^{(3)} \text{ is a constant.}
$$

This means at each fixed point $\frac{k}{T}$, the difference between $Q_i(\gamma_i)$ and $Q_i(\hat{\gamma}_i)$ is bounded by $M_{-1}^{(3)} \varepsilon$, and it is straightforward to have

$$
\| Q_i(\gamma_i) - Q_i(\hat{\gamma}_i) \|_2 \leq \sqrt{T} M_{-1}^3 \varepsilon.
$$

We finally get

$$
\Delta Q_{\text{reg}}(\gamma^*, \hat{\gamma}) = |Q_{\text{reg}}(\gamma^*) - Q_{\text{reg}}(\hat{\gamma})|
$$

$$
= \left| \sum_{j=1}^{C} \frac{1}{N^{(j)}} \sum_{i \in \{y_i = j\}} \| Q_i(\gamma_i^*) - \overline{Q}^{(j)} \|^2 - \sum_{j=1}^{C} \frac{1}{N^{(j)}} \sum_{i \in \{y_i = j\}} \| Q_i(\hat{\gamma}_i) - \overline{\hat{Q}}^{(j)} \|^2 \right|
$$

$$
\leq \sum_{j=1}^{C} \frac{1}{N^{(j)}} \sum_{i \in \{y_i = j\}} \left| \| Q_i(\gamma_i^*) - \overline{Q}^{(j)} \|^2 - \| Q_i(\hat{\gamma}_i) - \overline{\hat{Q}}^{(j)} \|^2 \right|.
$$

We now simplify $\frac{1}{N^{(j)}} \sum_{i \in \{y_i = j\}} \left| \| Q_i(\gamma_i^*) - \overline{Q}^{(j)} \|^2 - \| Q_i(\hat{\gamma}_i) - \overline{\hat{Q}}^{(j)} \|^2 \right|$ as

$$
\frac{1}{N^{(j)}} \sum_{i \in \{y_i = j\}} \left| \| a_i \|_2^2 + \| c \|_2^2 + 2\langle a_i, b_i \rangle + 2\langle b_i, c \rangle + 2\langle a_i, c \rangle \right|,
$$

where $a_i = Q_i(\gamma_i) - Q_i(\hat{\gamma}_i)$, $b_i = Q_i(\hat{\gamma}_i) - \overline{Q}^{(j)}$, $c = \overline{Q}^{(j)} - \overline{\hat{Q}}^{(j)}$.

By the Cauchy-Schwarz inequality: $\langle x, y \rangle \leq \|x\|_2 \|y\|_2$, we have $\|a_i\|_2 \leq \sqrt{T} M_{-1}^{(3)} \varepsilon$, and $\|c\|_2 = \left\| \frac{1}{N^{(j)}} \sum_{i \in \{y_i=j\}} a_i \right\|_2 \leq \sqrt{T} M_{-1}^{(3)} \varepsilon$. We also have $\|b_i\|_2 \leq M^{(4)}$ due to the boundness of $q_i$ and $\gamma_i$.

Thus, we have

$$\left| \frac{1}{N^{(j)}} \sum_{i \in \{y_i=j\}} \|Q_i(\gamma_i) - \overline{Q}^{(j)}\|_2^2 - \frac{1}{N^{(j)}} \sum_{i \in \{y_i=j\}} \|Q_i(\hat{\gamma}_i) - \overline{\hat{Q}}^{(j)}\|_2^2 \right| \leq 4T \left( M_{-1}^{(3)} \right)^2 \varepsilon^2 + 4\sqrt{T} M_{-1}^3 M^{(4)} \varepsilon.$$

Therefore, it is straightforward to have

$$\Delta Q_{\text{reg}}(\gamma^*, \hat{\gamma}) \leq 4CT \left( M_{-1}^{(3)} \right)^2 \varepsilon^2 + 4C\sqrt{T} M_{-1}^{(3)} M^{(4)} \varepsilon.$$

## D    PROOF OF THEOREM 3.3

Based on Theorem 2 from Yao et al. (2021), we only need to verify that the loss function and its gradient are both Lipschitz continuous. Since the mean SRVF ($\tilde{Q}^{(j)}$'s) are treated as constants when computing their gradients, and we have the condition that there exists $\epsilon_0 > 0$, such that $|\bar{Q}^{(u)} - \bar{Q}^{(v)}| \geq \epsilon_0$ for $1 \leq u < v \leq C$, we can omit the term $\sum_{1 \leq u < v \leq C} \alpha \|\bar{Q}^{(u)} - \bar{Q}^{(v)}\|^{-1}$, and simplify the loss function of DeepFRC for any individual as

$$l(\hat{f}_\Theta(\boldsymbol{x}_i), y_i) = \sum_{j=1}^{C} \|Q_i - \overline{Q}^{(j)}\| \mathbb{1}_{\{i \in \{y_i=j\}\}} - \sum_{j=1}^{C} y_{ij} \log \psi_{ij}.$$

The loss function is obviously continuous for $\psi_{ij} \geq \epsilon_0$, as $\frac{1}{z}$ is continuous everywhere except at zero. Similar to Eqs. (A.1) and (A.2), the gradient of $l$ with respect to any $\theta \in \Theta$ can be easily derived. These gradients can be expressed as products of gradients from ReLU, max pooling, etc., which are Lipschitz continuous. We note that both the 1D linear interpolation and a Fourier basis expansion in the stage of spectral representation in DeepFRC satisfy the Lipschitz continuity condition. Therefore, the theorem follows directly from Theorem 3.8 in Hardt et al. (2016).

## E    EXPERIMENTAL DETAILS

### E.1    EVALUATION METRICS

**Registration Metrics:**

• Registration Error ($\Delta Q_{\text{reg}}$): This metric, defined within the EFDA framework, quantifies the discrepancy between the estimated warping $\hat{\gamma}$ and the ground-truth $\gamma^*$. It is applicable only to simulated datasets where $\gamma^*$ is known. Lower values indicate better alignment.

• Adjusted Total Variance (ATV): For both simulated and real-world data, we use ATV to assess alignment without ground-truth warping functions. It is defined as: $\text{ATV} = \frac{1}{\binom{C}{2}} \sum_{1 \leq u < v \leq C} \frac{\text{TV}_{uv}}{d(\text{mean}(\tilde{x}_u), \text{mean}(\tilde{x}_v))}$, where $\text{TV}_{uv}$ (Chen & Srivastava, 2021) is the total variation between classes $u$ and $v$, and $\text{mean}(\tilde{x})$ is the mean function of the aligned curves. ATV accounts for inter-class distance, making it suitable for evaluating alignment of similar classes. Lower ATV indicates better alignment.

**Reconstruction Metric:**

• Coefficient Correlation ($\rho$): To evaluate the recovery of the underlying smooth process, we compute Pearson's correlation $\rho(\boldsymbol{c}^*, \hat{\boldsymbol{c}})$ between the true ($\boldsymbol{c}^*$) and estimated ($\hat{\boldsymbol{c}}$) Fourier basis coefficients. This metric is used only in simulations where $\boldsymbol{c}^*$ is accessible. Higher values indicate better reconstruction.

**Classification Metrics:**

• Accuracy (ACC): The proportion of correctly classified samples.

• Macro-averaged $F_1$-score: Provides a balanced measure of performance across classes, especially important for imbalanced datasets (Sokolova & Lapalme, 2009).

Higher values for both ACC and $F_1$-score indicate superior classification performance.

### E.2 DESCRIPTION OF SIMULATED DATASETS

We construct a balanced two-class functional dataset $\{x_i(t), y_i\}_{i=1}^N$, where each function $x_i(t)$ for $t \in [0, 1]$ is generated by composing amplitude and phase components. The amplitude function is defined as the sum of two Gaussian bumps:

$$z_i(t \,|\, a_i, \mu, \sigma_i) = a_i \cdot \exp\left(-\frac{1}{2}\frac{(t-\mu)^2}{\sigma_i^2}\right),$$

where $a_i$ and $\sigma_i$ are independently drawn from uniform distributions, and $\mu$ is a fixed constant. To introduce phase variation, we define a warping function $\gamma_i(t)$ as

$$\gamma_i(t) = \begin{cases} \frac{\exp(b_i t)-1}{\exp(b_i)-1}, & \text{if } b_i \neq 0, \\ t, & \text{if } b_i = 0, \end{cases} \quad \text{with } b_i \sim U(-1.5, 1.5).$$

Such constructions are widely used for modeling functional data with phase variability (Kneip & Ramsay, 2008; Tucker et al., 2013). We then generate each function as:

$$x_i(t) = \begin{cases} \sum_{j=1}^2 z_i(\gamma_i(t) \,|\, a_i^{(0j)}, \mu^{(0j)}, \sigma_i^{(0j)}), & \text{if } y_i = 0, \\ \sum_{j=1}^2 z_i(\gamma_i(t) \,|\, a_i^{(1j)}, \mu^{(1j)}, \sigma_i^{(1j)}), & \text{if } y_i = 1, \end{cases}$$

where the parameters $\{a_i^{(cj)}, \mu^{(cj)}, \sigma_i^{(cj)}\}_{j=1}^2$ for $c = 0, 1$ are summarized in in the following table. We simulate $N = 6000$ functions, each evaluated at 1000 time points. We split the balanced dataset into training, validation, and test sets with sizes 1600, 400, and 4000, respectively.

Table A1: Parameter setting for simulated data

| Index | (01) | (02) | (11) | (12) |
|---|---|---|---|---|
| $a_i$ | $13 + U(-0.5, 0.5)$ | $12.5 + U(-1, 1)$ | $12 + U(-1, 1)$ | $13 + U(-1.5, 1.5)$ |
| $\mu$ | 0.250 | 0.715 | 0.225 | 0.695 |
| $\sigma_i$ | $0.06 + U(-0.003, 0.003)$ | $0.075 + U(-0.003, 0.003)$ | $0.06 + U(-0.003, 0.003)$ | $0.1 + U(-0.003, 0.003)$ |

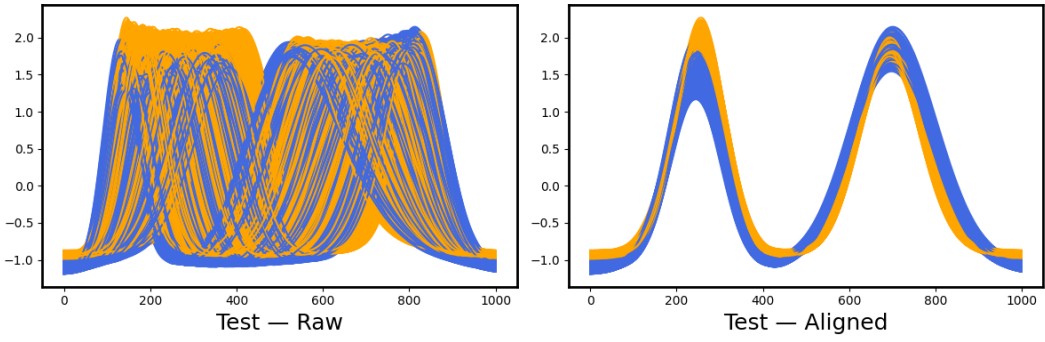

Figure A1: Visualization of alignment by DeepFRC on simulated test data, with evaluation metrics $\rho = 0.976$, $\Delta Q_{\text{reg}} = 0.056$, ATV=0.504, ACC $= 1.000$ and $F_1$-score $= 1.000$.

### E.3 DESCRIPTION OF REAL DATASETS

The first four time series datasets are one-dimensional functional data, from the UCR Time Series Classification Archive (www.timeseriesclassification.com), and the last dataset is three-dimensional functional data, from Kaggle (www.kaggle.com):

The Wave dataset (Liu & Yang, 2009) consists of eight simple gestures generated using accelerometers, collected through a particular procedure. For a participant, gestures are gathered when they hold a device and repeat certain gestures multiple times during a time period. The dataset includes X, Y, and Z dimensions with 8 classes. Here, we selected the X dimension and classes 2 and 8 for the

experiment. There are 1120 samples with label partition 591/529, and the train/validation/test split is 320/80/720, with each sample having 315 time points.

The Yoga dataset consists of images of two actors (one male, one female) transitioning between yoga poses in front of a green screen. The task is to classify the images based on the actor's gender. Each image was transformed into a one-dimensional series by measuring the distance from the outline of the actor to the center. The dataset contains 3300 samples with label partition 1770/1530, and the train/validation/test split is 800/200/2300. Each sample has 426 time points.

The Symbol dataset involves 13 participants, who were asked to replicate a randomly appearing symbol. There were 3 possible symbols, creating a total of 6 classes, and each participant made approximately 30 attempts. The dataset contains X-axis motion data recorded during the process of drawing the shapes. We conduct experiments on both the 2-class and 3-class cases using the Symbol dataset. In the 2-class case, there are 343 samples with label partition 182/167, and the train/validation/test split is 115/28/200. In the 3-class case, there are 510 samples with label partition 182/167/161, and the train/validation/test split is 168/42/300. Each sample has 398 time points.

The MotionSense (Malekzadeh et al., 2019) dataset involves multivariate time-series signals recorded from smartphone sensors during six daily activities performed by 24 participants. From this dataset, we selected three signal channels (G.x, G.y and G.z) from two actions (walk and jog) to construct a binary classification task, containing 96 samples with 200 time points each. The train/validation/test split is 60/16/20.

### E.4 INTRODUCTION OF BASELINES

Here we provide a brief introduction of the compared baseline methods:

• TTN (Lohit et al., 2019): A method for joint alignment and classification of discrete time series but does not account for curve smoothness.

• SrvfRegNet (Chen & Srivastava, 2021): An unsupervised method that aligns raw functional data via a 1D CNN.

• $FCNN_{raw}$: A model that discretizes raw functional data into a vector, inputting it into a fully connected neural network.

• $FCNN_{fourier}$: A model that transforms the functional data into Fourier basis coefficients and then feeds the resulting vector into a fully connected neural network.

• FuncNN (Thind et al., 2020): A model that inputs entire functional curves directly into a fully connected neural network.

• ADAFNN (Yao et al., 2021): A model that inputs entire functional curves, adapting the bases during learning.

• TSLANet (Eldele et al., 2024): A model that inputs entire functional curves, with ASB (Adaptive Spectral Blocks) and ICB (Interactive Convolutional Blocks) as core structure.

### E.5 MODEL IMPLEMENTATION DETAILS ON REAL DATA

For all real-world datasets, the functional input is Z-score standardized entry-wise based on the mean function and standard deviation. To handle missing values (such as NAs) in the functional input, we use the `BasisSmoother` function from the `skfda.preprocessing.smoothing` module in `Python` to impute missing values. All of these baseline models are trained until the best performance is achieved. All the experiments are conducted on NVIDIA GeForce RTX 3090 32G. The details of implementation for all models are provided as follows:

• The `DeepFRC` network is implemented as a `PyTorch` neural network. The hyperparameter settings for DeepFRC are shown in Table A2. In the discussion of stability with respect to basis expansion for `DeepFRC`, we compute Fourier, B-spline, and Polynomial scores as follows: The Fourier scores are obtained using the fast Fourier transform function `rfft` from `torch.nn.fft`, extracting the first 100 scores. The B-spline scores are derived from the cubic B-spline basis with 98 knots uniformly distributed over the interval $[0, 1]$, resulting in exactly 100 B-spline basis functions. The Polynomial scores are obtained using the Chebyshev polynomial orthogonal basis, taking the

Table A2: Hyperparameter setting for DeepFRC on four real datasets

| Case | $\alpha$ | $\beta$ | $lr_{\text{reg}}$ | $lr_{\text{class}}$ |
|---|---|---|---|---|
| Wave | 3000 | 50 | 1e-3 | 1e-3 |
| Yoga | 600 | 15 | 1e-3 | 5e-4 |
| Symbol (2 classes) | 300 | 15 | 1e-3 | 1e-3 |
| Symbol (3 classes) | 10 | 10 | 1e-3 | 1e-3 |
| MotionSense | 15 | 1 | 1e-3 | 1e-3 |

first 100 Chebyshev polynomial functions. The B-spline and Polynomial scores are both calculated using the `BasisSmoother` function from the `skfda.preprocessing.smoothing` module in `Python`.

• The model `SrvfRegNet` processes functional data through a learnable pre-warping block with three 1D convolutional layers (16, 32, 64 channels, kernel size 3, ReLU/BatchNorm/pooling), followed by a fully connected linear layer to generate warping functions. It applies time warping by minimizing the SRVF loss under an MSE criterion. The input data is the full functional data requiring alignment. The output of this network is the time-warping function that can generate aligned functional data. We use the default architecture for the models `SrvfRegNet`.

• Both $\text{FCNN}_{\text{raw}}$ and $\text{FCNN}_{\text{fourier}}$ have the same architecture: a 3-layer MLP, with the input layer consisting of $n$ observations or 100 Fourier scores (by default), followed by hidden layers with dimensions [16, 8], and finally outputting the classification results. Each layer is equipped with Layer Normalization, the ReLU activation function, and a dropout rate of 0.1. The input of $\text{FCNN}_{\text{raw}}$ is the raw data, while the input of $\text{FCNN}_{\text{fourier}}$ is derived from the first 100 Fourier coefficients of the functional data.

• The models `FuncNN` and `ADAFNN` both take the entire curve as input. `FuncNN` uses a manually chosen basis expansion, while `ADAFNN` employs an adaptively learned basis layer for representing curve data. We use the default architecture for the models `FuncNN` and `ADAFNN`.

• The model `TSLANet` takes univariate or multivariate time series as input, first processing them via patch segmentation and positional embedding to retain temporal order. Its core structure consists of Adaptive Spectral Blocks (for dependency capture and denoising) and Interactive Convolutional Blocks. We use the default architecture for the model `TSLANet`.

• A note on handling multidimensional functional data is necessary. Models including `DeepFRC`, `TTN`, `TSLANet`, and `FuncNN` natively support such inputs. For other baselines, we implemented the following adaptations: `SrvfRegNet` was extended to process multidimensional data analogously to `DeepFRC`. For $\text{FCNN}_{\text{raw}}$ and `ADAFNN`, we concatenated signals from all channels into a single one-dimensional vector. For $\text{FCNN}_{\text{fourier}}$, we used the concatenated basis expansion coefficients from each channel as the model input.

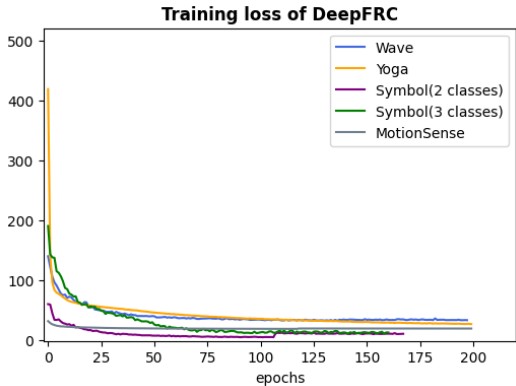

Figure A2: Training loss vs. epochs by DeepFRC across the five real-life datasets.

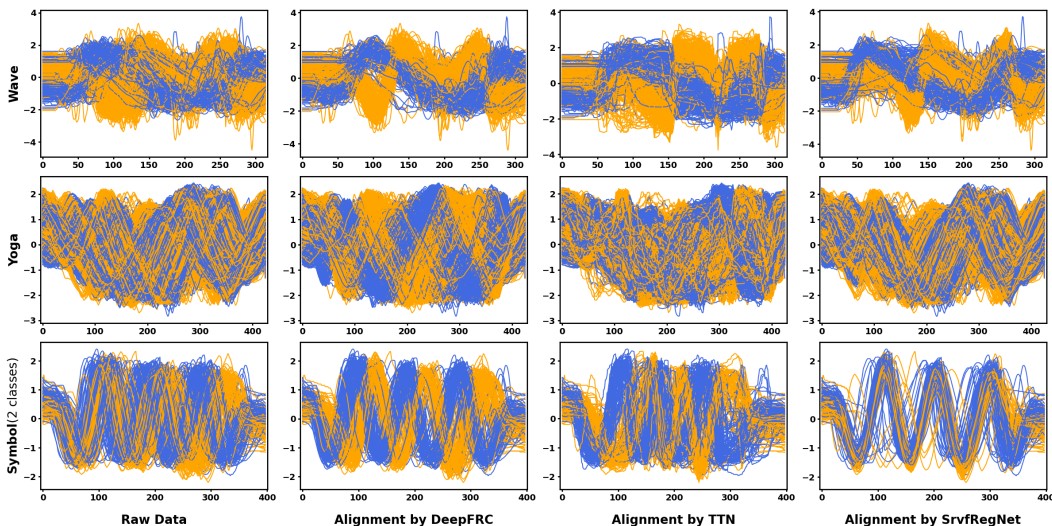

Figure A3: Visualization comparison of alignment performance by DeepFRC, TTN and SrvfRegNet across three two-class real datasets

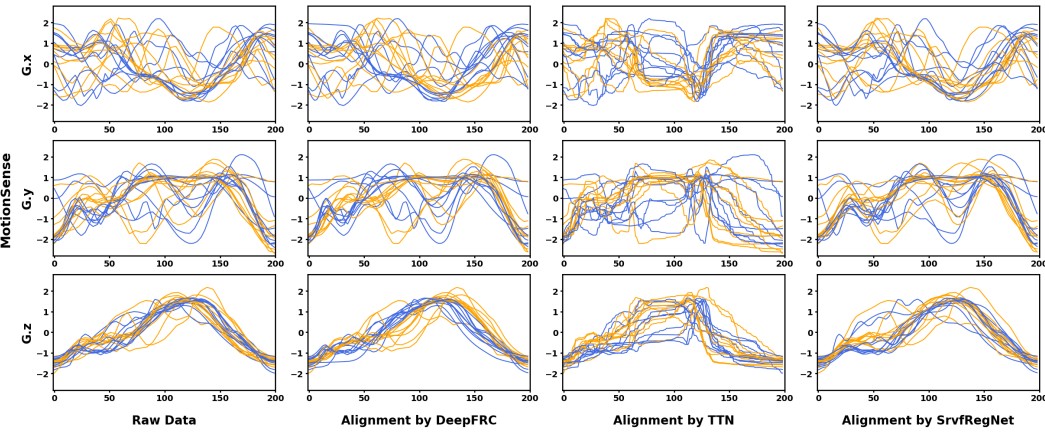

Figure A4: Visualization comparison of alignment performance by DeepFRC, TTN and SrvfRegNet on MotionSense dataset in 3 channels G.x, G.y and G.z.

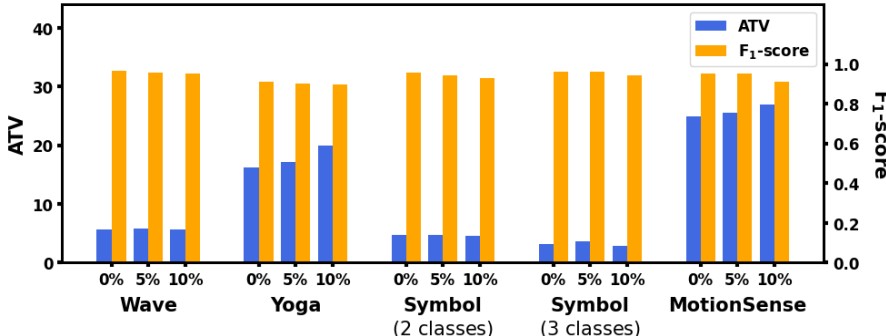

Figure A5: Performance of DeepFRC with different missing rates for raw data across five real-world datasets.

Table A3: Representation coefficients correlation on different noise level on synthetic data

| Model | Noise level $\varepsilon \sim N(0, \sigma^2)$ | | | |
|---|---|---|---|---|
| | $\sigma = 0$ | $\sigma = 0.05$ | $\sigma = 0.10$ | $\sigma = 0.20$ |
| DeepFRC | 0.983 | 0.982 | 0.979 | 0.969 |
| TTN | 0.541 | 0.529 | 0.501 | 0.490 |
| SrvfRegNet | 0.969 | 0.966 | 0.961 | 0.950 |

Table A4: Quantitative comparison of registration and classification performance on large scale real dataset: 2-class Symbol dataset ($100\times$). **Bold** indicates best results.

| Model | ATV | ACC | $F_1$-score |
|---|---|---|---|
| DeepFRC | **4.8** | **94.47%** | **0.944** |
| TTN | 6.4 | 90.46% | 0.897 |
| SrvfRegNet+FCNN$_{raw}$ | 12.8 | 93.29% | 0.933 |
| SrvfRegNet+FCNN$_{fourier}$ | 12.8 | 93.08% | 0.930 |
| SrvfRegNet+FuncNN | 12.8 | 92.23% | 0.921 |
| SrvfRegNet+ADAFNN | 12.8 | 90.08% | 0.942 |
| SrvfRegNet+TSLANet | 12.8 | 94.02% | 0.940 |

Table A5: Robustness under data scarcity. $N$ and $n$ represent sample size and sequence length, respectively.

| Evaluation | Sparse samples ($n = 1000$) | | | Sparse time points ($N = 6000$) | | | Sparse samples and time points $(N, n)$ | | |
|---|---|---|---|---|---|---|---|---|---|
| | $N = 200$ | $N = 100$ | $N = 50$ | $n = 200$ | $n = 100$ | $n = 50$ | $(200, 200)$ | $(100, 100)$ | $(50, 50)$ |
| ATV | 0.580 | 0.784 | 0.867 | 0.605 | 2.105 | 2.571 | 1.097 | 2.454 | 12.492 |
| ACC | 100.0% | 100.0% | 100.0% | 100.0% | 99.3% | 98.2% | 100.0% | 95.0% | 80.0% |
| $F_1$-score | 1.000 | 1.000 | 1.000 | 1.000 | 0.992 | 0.980 | 1.000 | 0.947 | 0.750 |

Table A6: Robustness to non-diffeomorphic warpings on synthetic data. *prob*: proportion of curves that are warped using functions with flat regions (zero derivative). *flat_frac*: flat fraction of the domain.

| Evaluation | Original version | $prob = 10\%$ | | | $prob = 20\%$ | | |
|---|---|---|---|---|---|---|---|
| | | *flat_frac* = 0.2 | *flat_frac* = 0.4 | *flat_frac* = 0.6 | *flat_frac* = 0.2 | *flat_frac* = 0.4 | *flat_frac* = 0.6 |
| ATV | 0.50 | 6.82 | 13.91 | 16.57 | 14.72 | 25.12 | 32.15 |
| $\rho$ | 0.976 | 0.940 | 0.920 | 0.911 | 0.881 | 0.847 | 0.828 |
| ACC | 100.00% | 95.65% | 95.25% | 94.15% | 93.45% | 93.05% | 90.43% |
| $F_1$-score | 1.000 | 0.958 | 0.952 | 0.943 | 0.935 | 0.932 | 0.913 |

