# OpenReview forum: "DeepFRC: An End-to-End Deep Learning Model for Functional Registration and Classification"
_ICLR.cc/2026/Conference — ICLR 2026 Poster_

### Official Review · Reviewer_gJ51 · 2025-10-28

**Soundness:** 3
**Presentation:** 3
**Contribution:** 3
**Rating:** 6
**Confidence:** 3

**Summary:**

The paper introduces a novel end-to-end deep learning model DeepFRC that integrates deformable registration with classification, tasks that are so far adressed usually separately. This is achieved utilizing a diffeomorphic neural registration operation, Fourier spectal representation for smooth functional encoding and a class-aware contrastive objective. Experiments on both synthetic and real datasets demonstrate consistently improved alignment together with competitive or superior classification performance.

**Strengths:**

1. Offers and end-to-end method to jointly learn functional registration and classification in a unified architecture rather than sequentially.
2. Theoretical guarantees are provided that limk registration fidelity to classification generalization.
3. Improved performance across real world and synthetic dataset and comprehensive ablation for each architectural component.
4. Robustness is shown against noise and missing data.
5. The language of the paper is clear.
6. Git link to code is available.

**Weaknesses:**

**Weaknesses and Questions**

1. Since TTN is also addressing jointly the registration and classification tasks, can the authors elaborate and highlight a bit more the architectural and conceptual differences between these methods?
2. How sensitive is the model to mis-specified diffeomorphic constraints if real warpings violate the diffeomorphic guarantees?
3. In line 082: the paper claims that the paper of Tang et al. is heavily reliant on assumptions,…
Can the authors clarify which assumptions those are and whether they avoid making these assumptions in this work?
4. In the experiments the paper claims that simulated data provide insights regarding registration, classification and reconstruction. I would like to understand why is it important to draw conclusions on the reconstruction along with the other 2 tasks discussed thoroughly in the paper.
5. In table 1 we sometimes see that several methods demonstrate high acc and F1 score while their registration performance is not optimal. Is there any intuition behind this? Does it mean that the SrvfRegNet was not tuned properly? Does this not make those methods robust against misregistration which can be an interesting and useful property? What is the intuition behind SrvfRegNet always not being able to recover the registration as accurately as the proposed method?
6. Another question regarding the results is whether there is any intuition why in the case of Symbol 3 the SrvfRegNet + TSLANet is delivering higher classification accuracy compared to Symbol2. Is there any correlation to the number of classes?
7. I would like to request some addition of limitations of the current method and of why one should not consider to addressing classification and registration jointly.

**Questions:**

Please see above.

If the authors address some of the discussion points above I am willing to increase my score in the rebuttal.

---

> ### Author Response · Authors · 2025-11-19
> **Response to Reviewer gJ51**
>
> We thank the reviewer for the thoughtful and detailed feedback. We have revised the manuscript accordingly, with all changes highlighted in red. Below we provide point-by-point responses.
>
> Q1: "Since TTN also performs joint registration and classification, how does it differ from DeepFRC?"
>
> We appreciate the question. Although both TTN and DeepFRC are joint models, they are built on fundamentally different principles. TTN targets generic discrete time series and learns warpings without enforcing smoothness or invertibility, effectively treating alignment as an unconstrained distortion—often yielding visibly distorted class trajectories (Fig.3(c)). In contrast, DeepFRC is designed for functional data: it enforces diffeomorphic, smooth warpings via a neural deformation operator and uses a spectral representation rooted in elastic functional data analysis. This leads to coherent, class-separated alignments (Fig.3(b)). DeepFRC further introduces a class-aware contrastive alignment loss and offers theoretical guarantees for both registration and generalization—features absent in TTN.
>
> Q2: "How sensitive is DeepFRC if real warpings violate diffeomorphic assumptions?"
>
> To assess robustness under misspecification, we generated additional synthetic test sets where a proportion of curves were warped using maps containing flat regions ($\gamma'(t)=0$) over a fraction of the domain, mimicking temporal "jumps". Results (Section 5) show:
> (i) Gradual degradation: even with 20\% of curves containing 60\% flat regions, accuracy remains 90.43\%.
> (ii) Alignment more affected than classification: ATV declines more than ACC/F1, but spectral features remain discriminative.
> (iii) Stable reconstruction: coefficient correlations remain high ($\rho>0.82$).
> Thus, while DeepFRC is designed for diffeomorphic warpings, it learns smooth approximations when violations occur and preserves discriminative information. These results are now included in Section 5.
>
> Q3: "Which assumptions in Tang et al. are avoided by DeepFRC?"
>
> Tang et al. rely on strong assumptions: (i) parametric warping models (e.g., B-splines with random effects), (ii) Gaussian process assumptions for both functions and random effects, and (iii) a two-stage alternating optimization prone to local minima. DeepFRC avoids these by (a) learning warpings nonparametrically, (b) making no distributional assumptions, and (c) using end-to-end optimization. We have clarified this in Section 1.2.
>
> Q4: "Why evaluate reconstruction in simulated experiments?"
>
> Reconstruction quality (via coefficient correlation $\rho$) is an essential diagnostic in simulations for three reasons: (i) it verifies that alignment preserves amplitude structure rather than distorting signals; (ii) it detects over-warping, which can degrade generalization; and (iii) it assesses the expressiveness of the spectral basis. Although real datasets lack ground-truth amplitudes, reconstruction metrics in simulations provide important validation that the model performs faithful functional alignment. We added clarification in Section 4.
>
> Q5: "Why do some methods achieve high ACC/F1 but poor registration? Why does SrvfRegNet underperform?"
>
> Two factors explain this observation: (i) Feature redundancy: strong classifiers such as TSLANet extract amplitude-driven features that remain robust even when registration is imperfect, yielding high ACC/F1. (ii) Dataset characteristics: Wave and Symbol contain amplitude-dominant class differences, making misregistration less harmful. SrvfRegNet underperforms in registration because it is fully unsupervised, aligning curves to a population mean rather than to class-specific structures. DeepFRC, guided by a class-aware contrastive loss, explicitly promotes class-separated alignments. We agree that robustness to misregistration is valuable and have expanded this discussion in Section 4.2.
>
> Q6: "Why does SrvfRegNet+TSLANet perform better on Symbol(3) than Symbol(2)?"
>
> SrvfRegNet+TSLANet achieves higher accuracy on Symbol(3) because its three classes are more distinct in amplitude space, enabling TSLANet to discriminate effectively despite misalignment. Symbol(2) contains classes with more similar amplitude patterns, making phase variability more influential and highlighting the advantages of DeepFRC. This matches visual patterns in  Fig.3 and Fig.A3.
>
> Q7:  "Please discuss limitations and when joint modeling may be unnecessary."
>
> We added a dedicated limitations paragraph in Section 6. DeepFRC may face challenges with (i) highly irregular or discontinuous trajectories that violate smoothness assumptions and (ii) substantial label noise, which can impair class-aware alignment. Moreover, joint registration--classification may be unnecessary when (a) phase variability is negligible, (b) labels are scarce or unreliable, or (c) a strong classifier already exhibits robust invariance to temporal misalignment. These considerations are now included in the revised conclusion.

---

### Official Review · Reviewer_UBjh · 2025-10-30

**Soundness:** 2
**Presentation:** 3
**Contribution:** 3
**Rating:** 6
**Confidence:** 3

**Summary:**

This paper introduces DeepFRC, an end-to-end deep learning framework that jointly addresses functional data registration (alignment) and classification. Its main contribution is a unified architecture that integrates a neural deformation operator for diffeomorphic warping, a spectral representation for smooth functional encoding, and a class-aware contrastive loss. This approach eliminates the need for separate preprocessing steps and allows the alignment and classification tasks to mutually enhance each other. The work is further supported by theoretical guarantees on its approximation capability and generalization error, and is empirically validated to outperform state-of-the-art methods on both synthetic and real-world datasets.

**Strengths:**

The paper demonstrates good originality by proposing the first end-to-end unified framework for joint functional data learning.
The research quality is solid, well-supported by comprehensive theoretical analysis and systematic experimental validation.

**Weaknesses:**

1. It is not clear about the network architecture choices.
2. Current computational complexity analysis is not sufficient.
3. Lack of a detailed description of the datasets, which helps to understand the possible applications of the proposed method.

**Questions:**

1. The paper uses a 1D CNN to parameterize the neural deformation operator (the registration module) but does not explain why Transformer-based models were not chosen. Can you explain the core reasons for this selection?
2. It is recommended to supplement the experimental results with statistical significance analysis to prove that the performance improvements are statistically significant.
3. The current computational efficiency comparison in the paper only mentions runtime. Would supplementing it with the total training time, model parameter count, and FLOPs provide a more comprehensive demonstration of DeepFRC's efficiency advantages?
4. Can you further clarify the sampling methods of the four real-world experimental datasets? Additionally, could you explain how the proposed method handles highly irregularly sampled functional data?
5. I would suggest adding more application analysis to demonstrate the effects of the proposed method.

---

> ### Author Response · Authors · 2025-11-19
> **Response to Reviewer UBjh**
>
> We thank the reviewer for the thoughtful and constructive feedback. We have revised the manuscript accordingly, with all changes highlighted in red. Below we provide point-by-point responses.
>
> Weakness 1 \& 3: "Clarity of architecture choices and dataset descriptions."
>
> We appreciate the comment. Full details of (i) the network architecture and (ii) all real-world datasets were already provided in the original submission:  (1) Architecture: Both Sections 2.1 \& 2.3 and Appendix B specify the 1D-CNN used in the neural deformation operator (4 layers, kernel size 3, channels 16$\rightarrow$32$\rightarrow$64, with ReLU, BatchNorm, MaxPooling) and the MLP classifier (3 layers with 8$\rightarrow$4 hidden units), together with motivations grounded in prior FDA literature.
> (2) Datasets: Appendix E.3 describes the Wave, Yoga, Symbol, and MotionSense datasets, including sample sizes, class balance, sequence lengths, and sensing modalities.
>
> Weakness 2: "Insufficient complexity analysis."
>
> We address this concern in Question 3 below, where we significantly expand the computational efficiency analysis.
>
> Question 1: "Why a 1D CNN rather than a Transformer for the deformation operator?"
>
> We thank the reviewer for this excellent question. A 1D CNN is chosen for three principled reasons:
> (1) Locality \& Smoothness: Warping functions $\gamma(t)$ are smooth and determined primarily by local neighborhoods. CNNs naturally encode locality and translation equivariance, making them well suited for smooth deformations, whereas long-range Transformer attention is unnecessary.
> (2) Computational Efficiency: Functional curves often have long sequences. CNNs scale as $\mathcal{O}(n)$, while Transformers incur $\mathcal{O}(n^2)$ complexity, which becomes prohibitive in end-to-end training.
> (3) Data Efficiency \& Regularization: CNNs use fewer parameters and provide implicit regularization, helping ensure the deformation operator outputs smooth, monotone warpings. Transformers risk overfitting and unstable warp generation.
> Thus, a 1D CNN aligns naturally with the mathematical structure of diffeomorphic registration and provides stable, efficient training.
>
> Question 2: "Statistical significance of improvements."
>
> We thank the reviewer for this suggestion. We have conducted paired $t$-tests across 10 random seeds to quantify the significance of the ablation results. Across all datasets, removing any core component leads to statistically significant degradation ($p<0.05$), with most cases showing strong significance ($p<0.01$). For example:
> • Removing the Neural Deformation Operator sharply reduces accuracy (MotionSense: $p=0.0000$; Symbol-3: $p=0.0000$).
> • Removing the Spectral Representation harms both alignment and classification (MotionSense ATV: $p=0.0025$; ACC: $p=0.0001$).
> • Removing classifier-feedback collapses alignment ($p=0.0000$ for all datasets).
> These results confirm that each component is essential. Section 4.2 now includes a summary, and the full $p$-value table is shown in Table 3.
>
> Question 3: "Expanded efficiency analysis."
>
> We appreciate the suggestion. Section 5 now includes:
> (1) Training Time: DeepFRC trains in $\sim$60s. (The previously reported “runtime’’ corresponded to training time; inference for all models is $\sim$1s.)
> (2) Model Size: DeepFRC has $\sim$200k parameters, comparable to other deep baselines.
> (3) FLOPs: Training requires $\sim$1000 TFLOPs, lower than SrvfRegNet-based hybrids (1500 TFLOPs) while delivering higher accuracy than TTN (200 TFLOPs).
> Combined with the $\mathcal{O}(Nn)$ theoretical complexity, these results demonstrate a favorable performance–efficiency trade-off.
>
> Question 4: "Sampling methods and irregularly sampled functional data."
>
> (1) Real-world sampling: Appendix E.3 in the original submission already details that Wave, Yoga, and Symbol (UCR datasets) are regularly sampled, and MotionSense is collected at a uniform 50~Hz.
> (2) Irregular sampling: DeepFRC naturally accommodates irregular observations: (a) inputs are general $(t_{ij}, x_{ij})$ pairs; (b) the deformation operator learns a continuous $\gamma_i(t)$ directly on these points; (c) a differentiable interpolation step (Eq. 3) maps warped curves to a common grid for spectral encoding. This enables application to biomedical and ecological data with heterogeneous sampling.
>
> Question 5: "Request for more application analysis."
>
> We agree that practical interpretation is valuable. While page limits prevent full case studies, Section 4 now includes illustrative examples:
> • MotionSense: DeepFRC synchronizes biomechanical events (heel strikes, mid-swings), producing physiologically coherent activity profiles.
> • Symbol: The model corrects for user-specific writing speeds and stroke orders, revealing canonical symbol shapes.
> These examples demonstrate that DeepFRC provides not only improved accuracy but also interpretable, domain-relevant functional templates.

---

> > ### Comment · Reviewer_UBjh · 2025-11-26
> > **Reviewer Response to Author Rebuttal**
> >
> > I would like to thank the authors for providing additional explanations regarding the choice of architecture and for including further experiments on statistical significance analysis. Regarding the application analysis, while the authors have added a paragraph for clarification, the potential applications of the method remain somewhat unclear to me. Given that the task addressed in this paper is not widely studied in the registration domain, I believe it would be beneficial to include a downstream experiment to better demonstrate the practical utility of the proposed method. Considering the overall contribution to the field, I maintain my previous assessment score.

---

> > > ### Author Response · Authors · 2025-11-26
> > >
> > > We sincerely thank the reviewer for the continued engagement with our work and for the encouraging remarks regarding the clarifications on architecture and the new statistical significance analyses. We are pleased that the revisions resolved most of the reviewer’s earlier concerns.
> > >
> > > Regarding the remaining point on application analysis, we fully agree that demonstrating downstream utility is important. At the same time, we would like to clarify the broader statistical context of our contribution. Functional registration (curve alignment) is a long-standing and central problem in functional data analysis (FDA), with extensive literature in top-tier statistics journals (e.g., Annals of Statistics, Biometrika, JRSS-B) showing its importance for extracting meaningful structure from temporally misaligned trajectories. Our paper addresses a fundamental FDA task: jointly improving registration and classification. The main practical contribution of DeepFRC is to show—both theoretically and empirically—that accurate class-aware registration can substantially enhance functional classification performance, a key downstream task in FDA applications ranging from biomechanics to biomedical signal analysis.
> > >
> > > We appreciate the suggestion to provide additional large-scale or domain-specific downstream case studies. However, we note that (i) the ICLR page limit already constrains the scope of empirical analyses, and (ii) our focus in this work is methodological: establishing an end-to-end framework with theoretical guarantees and demonstrating across multiple benchmark datasets that improved alignment directly translates into better classification, interpretability, and robustness. Adding further downstream applications (e.g., forecasting, clustering, or domain-specific clinical analyses) would require substantial additional domain-specific modeling and exceed the intended scope of this paper.
> > >
> > > We nevertheless believe the revised manuscript now clearly communicates the practical utility of DeepFRC: it provides interpretable class-specific templates, aligns heterogeneous trajectories in a statistically principled manner, and improves classification accuracy across a diverse suite of real-world datasets. These contributions advance the FDA literature by integrating two essential and traditionally decoupled tasks into a unified, theoretically grounded deep learning architecture.
> > >
> > > We thank the reviewer again for the thoughtful feedback and hope that the clarified context and expanded empirical and theoretical analyses help convey the significance and practical relevance of our work.

---

### Official Review · Reviewer_q1ar · 2025-10-30

**Soundness:** 3
**Presentation:** 2
**Contribution:** 3
**Rating:** 4
**Confidence:** 4

**Summary:**

The paper proposes DeepFRC, an end-to-end framework for function/sequence alignment and representation-driven registration/classification. The method builds a deformation operator for time warping, a spectral/spectral-coefficient representation of aligned functions, and a classifier that uses contrastive geometric alignment objectives.

**Strengths:**

The authors provide theoretical guarantees, proving that the model can approximate optimal warping functions and establishing a data-dependent generalization bound that links registration fidelity to classification performance.

**Weaknesses:**

1.	The authors claim “but rarely addressing both simultaneously”, however, there are several works addressing registration and classification simultaneously, for example,
[1] Zhang, Y. and Telesca, D., 2014. Joint clustering and registration of functional data. arXiv preprint arXiv:1403.7134.
2.	Why transform the latent features into a monotone cumulative sum can guarantee diffeomorphism?
3.	Novelty is limited, for registration, only introduce neural deformation operator for alignment
4.	The claim that DeepFRC is "efficient" is poorly supported. The authors only report inference time, which is already longer than other learning-based methods. Crucially, they omit training time, computational complexity analysis, and a comparison of model parameters (number of weights).
5.	The theoretical results rely on assumptions (e.g., Lipschitz continuity, compactness) that should be discussed more critically.
6.	The comparison methods are from 2021, the authors should compare most recent methods, like the methods published in 2025 and 2024.

**Questions:**

1. add more comparisons to current methods.
2. explain why transform the latent features into a monotone cumulative sum can guarantee diffeomorphism?

---

> ### Author Response · Authors · 2025-11-19
> **Response to Reviewer q1ar**
>
> We thank the reviewer for the constructive feedback and for the opportunity to improve the paper. All requested clarifications and revisions have been incorporated in the updated manuscript (highlighted in red). Below we provide point-by-point responses.
>
> Weakness 1: "Rarely addressing both simultaneously" vs.  Zhang \& Telesca (2014).
>
> We appreciate the reference. Zhang and Telesca (2014) proposed a Bayesian hierarchical model for joint clustering and registration—an unsupervised learning approach that is fundamentally different from our supervised classification setting. In contrast, our work tackles the distinct problem of jointly performing diffeomorphic registration and classification within an end-to-end deep learning framework. To the best of our knowledge, prior deep models (e.g., SrvfRegNet, ADAFNN, FuncNN, FCNN) treat registration and classification as separate tasks. We clarified this distinction in Section 1.2.
>
> Weakness 2 / Question 2: "Why does monotone cumulative summation guarantee a diffeomorphism?"
>
> As described in Section 2.1, the neural deformation operator outputs increments $\tau_i$, which we square to enforce $\tau_i^2 \ge 0$, followed by cumulative summation and normalization to satisfy boundary conditions. This ensures: (i) strict monotonicity ($\gamma(t_{i+1}) > \gamma(t_i)$), (ii) invertibility on a compact interval, (iii) smoothness because the CNN produces a continuous $\tau(t)$, and (iv) in the continuous limit, $\gamma'(t)>0$, yielding a $C^1$ diffeomorphism. This construction is standard in neural registration (e.g., Chen \& Srivastava, 2021). We have expanded this explanation at the end of Section 2.1.
>
> Weakness 3: "Novelty is limited---only a neural deformation operator is introduced."
>
> We respectfully disagree. The contribution of DeepFRC is not a single module but the first unified, end-to-end framework for joint diffeomorphic registration and supervised classification of functional data. Its novelty arises from the synergy of: (1) a diffeomorphic neural deformation operator, (2) a smooth spectral representation tailored to functional data, (3) a class-aware contrastive alignment loss (Eq.7) that explicitly couples registration fidelity with class separation, and (4) a theoretical framework (Theorems 3.1 and 3.3) linking registration error to generalization. Ablation results (Table 2) confirm that each component is essential. We  clarified this in Section 1.3.
>
> Weakness 4: "Efficiency claim is unsupported; inference time alone is insufficient."
>
> We thank the reviewer for noting this. The runtime reported in Section 5 is indeed training time (not inference time), and we corrected the terminology. Inference is nearly instantaneous ($\sim$1s) for all deep models. Section 2.5 already establishes the linear-time complexity $\mathcal{O}(Nn)$, which is significantly more efficient than quadratic-cost methods such as DTW. To substantiate our efficiency claims, we now include a more comprehensive computational analysis in Section 5. (1) Training FLOPs: DeepFRC requires approximately 1000 TFLOPs during training, which is substantially lower than SrvfRegNet-based hybrids (1500 TFLOPs) while remaining reasonable relative to lighter models such as TTN (200 TFLOPs). (2) Model size: DeepFRC contains only ~200k parameters, making it parameter-efficient and comparable to other deep baselines; its performance does not rely on excessive capacity. Together, these results show that DeepFRC delivers superior accuracy with competitive and well-justified computational costs, achieving a strong balance between performance and efficiency.
>
> Weakness 5: "Assumptions (Lipschitz, compactness) should be discussed more critically."
>
> We appreciate this suggestion. We have added a “Discussion of Assumptions” paragraph (end of Section 3) explaining how the Lipschitz and compactness assumptions are naturally satisfied through standard training practices (e.g., weight decay, Lipschitz activation mappings, and bounded spectral expansions). We further clarify what happens when these assumptions fail (e.g., $\gamma'(t)$ loses positivity, diffeomorphic guarantees break, and performance may degrade), tying the theoretical conditions directly to practical robustness considerations.
>
> Weakness 6 / Question 1: “Baselines are outdated; add 2024/2025 methods.”
>
> This appears to be a misunderstanding. We already include TSLANet (2024), one of the most recent and strongest time-series architectures, in Table 1. To the best of our knowledge, no deep-learning methods from 2024–2025 address either (i) joint functional registration + classification, or (ii) functional registration alone. We benchmark against all known functional registration baselines and strong recent classifiers for completeness. We remain happy to include any specific new method suggested by the reviewer.

---

> ### Author Response · Authors · 2025-12-01
> **Clarification of Factual Misunderstandings in Reviewer Comments**
>
> We thank the reviewer for the time and effort. We respectfully note that several of the stated weaknesses appear to stem from factual misunderstandings of the paper, which may have adversely influenced the evaluation. We summarize the key points below.
>
> (1) Misinterpretation of Prior Work
>
> The reviewer cites Zhang \& Telesca (2014), which concerns unsupervised clustering and registration. Our work focuses on supervised joint registration and classification in an end-to-end deep learning framework—a fundamentally different problem. All relevant prior work in this category is already cited.
>
> (2) Claim That Recent Baselines Are Missing
>
> The reviewer suggests that recent 2024–2025 models are not included. However, TSLANet (2024)—one of the strongest and most recent time-series models—is already evaluated throughout our experiments. To our knowledge, no 2024–2025 methods address the specific task of joint functional registration and classification.
>
> (3) Theoretical Assumptions Already Discussed
>
> The reviewer requests discussion of Lipschitz and compactness assumptions, but these conditions are already addressed at the end of Section 2.2 and in the proof of Theorem 3.3. We are happy to expand further, but the existing discussion appears to have been overlooked.
>
> (4) Confusion About Efficiency Metrics
>
> The reviewer interprets our runtime results as inference time; they are in fact training times. We have clarified this in the revision and provided additional computational analysis.
>
> We greatly appreciate constructive feedback and carefully addressed all reviewer comments. Given the number of factual misunderstandings, we respectfully ask the AC to consider this context when assessing the contribution and correctness of the submission.

---

### Official Review · Reviewer_PLqL · 2025-10-31

**Soundness:** 3
**Presentation:** 3
**Contribution:** 3
**Rating:** 6
**Confidence:** 3

**Summary:**

This paper presents DeepFRC, a deep learning framework that jointly performs functional registration and classification for functional or trajectory data. It integrates a neural deformation operator with 1D CNN to learn diffeomorphic time-warping functions for temporal alignment, a spectral representation module based on Fourier bases for smooth function embedding, and a classifier trained with a contrastive–geometric loss to align within classes and separate between classes.

Theoretical analysis shows that DeepFRC can approximate optimal warpings and achieves bounded generalization error. Experiments on both synthetic and real-world datasets (Wave, Yoga, Symbol, MotionSense) demonstrate improved alignment and classification accuracy over baselines such as TTN and SrvfRegNet.

**Strengths:**

1. The paper presents a well-motivated problem by targeting the joint challenge of phase variability and classification in functional data analysis (FDA).
2. Empirical results are consistent: DeepFRC outperforms alternatives across several datasets, enhancing both alignment and classification and confirming the effectiveness of joint optimization.
3. Theoretical discussions and included proofs, effectively situate the model within the mathematical landscape of FDA.

**Weaknesses:**

1. The neural components (1D CNN, MLP, Fourier basis) are standard. The main contribution is integrating known elements instead of developing a new architecture or loss function.
2. The baseline models are too few and outdated; it would be better to include more recent baseline models for comparison.

**Questions:**

1. Theorems 3.1 and 3.3 rely on many assumptions such as smoothness, bound and compactness. How realistic are these for real-world datasets with irregular sampling or noise?
2. How would the baseline models perform when scaling to the large datasets (time complexity, ATV, ACC, etc.)

---

> ### Author Response · Authors · 2025-11-19
> **Response to Reviewer PLqL**
>
> We thank the reviewer for the thoughtful and constructive feedback. We have revised the manuscript accordingly, with all changes highlighted in red. Below we provide point-by-point responses.
>
> Weakness 1: "The neural components are standard; the contribution is mainly integrative."
>
> We appreciate the reviewer’s perspective but respectfully disagree that the contribution is merely integrative. While DeepFRC leverages standard neural primitives for stability and efficiency, its core contributions are substantial:
> (1) A Novel Loss Function.
> We introduce a class-aware contrastive alignment loss (Eq. 7), which—unlike standard contrastive or triplet objectives—explicitly couples (i) intra-class SRVF variance (a registration criterion) with (ii) inter-class SRVF separation (a discriminative criterion). This is a new objective tailored to elastic functional data analysis and is central to enabling joint registration--classification.
> (2) A New Unified Architecture.
> DeepFRC is the first end-to-end model that jointly learns a diffeomorphic deformation operator and a classifier for functional data. Prior deep learning works are either (i) not joint, (ii) do not enforce diffeomorphic constraints, or (iii) are not designed for continuous functional observations.
> (3) Theoretical Guarantees.
> We provide the first theoretical results for a joint registration--classification model (Theorems 3.1 and 3.3), linking registration error to classification generalization.
> Thus, the main contributions extend beyond integration: DeepFRC introduces a new loss formulation, a unified architectural paradigm, and a theoretical framework that together enable a level of synergy between registration and classification not present in prior work.
>
> Weakness 2: "Baseline models are too few and outdated."
>
> We respectfully disagree. To the best of our knowledge, our comparisons include all relevant and most recent baselines:
> (1) Joint Registration+Classification. TTN is the only deep model attempting this joint task.
> (2) Registration. SrvfRegNet (2021) remains the state-of-the-art deep functional registration model.
> (3) Classification. We include TSLANet (2024), a very recent and strong time-series classifier outperforming many Transformer-based models, as well as ADAFNN (2021) and FuncNN (2020).
> To our knowledge, there are no 2024–2025 deep methods for joint registration and classification of functional data. TSLANet represents the strongest recent classification baseline. We are happy to add any specific additional baselines the reviewer recommends.
>
> Question 1: "How realistic are the smoothness/compactness assumptions in Theorems 3.1 and 3.3?"
>
> Thank you for this important question. The assumptions are standard in functional data analysis and are reasonable in practice:
> (1) Smoothness and Boundedness.
> Most real-world functional signals (e.g., biomechanics, sensor readings) are inherently smooth due to underlying physical processes. Preprocessing (e.g., mild smoothing, handling irregular sampling) and the SRVF representation further encourage smoothness. Our experiments include noise and missing data, and DeepFRC performs robustly even when these assumptions are not perfectly met.
> (2) Compactness.
> Weight normalization and weight decay constrain the parameter space during training, ensuring effective compactness. Modern neural architectures and bounded activations help preserve Lipschitz continuity.
> While extremely noisy or highly irregular data may nominally violate the assumptions, our empirical results (Section 5) show graceful performance degradation, indicating that DeepFRC generalizes well beyond ideal theoretical conditions.
>
> Question 2: "How do baseline models scale on large datasets?"
>
> We thank the reviewer for this question and have expanded the scalability study accordingly. We conducted a new large-scale experiment using a $100\times$-augmented Symbol dataset, evaluating all baselines (not only DeepFRC). Results are provided in Table A4 (Appendix E.5). The findings are:
> (1) Superior Performance at Scale.
> DeepFRC achieves the highest accuracy (94.47\%) and $F_1$-score (0.944), while also providing the best alignment quality (ATV $= 4.8$).
> (2) Efficiency.
> Consistent with its $\mathcal{O}(Nn)$ complexity (Section 2.5), DeepFRC trains efficiently and remains competitive with other deep baselines on large datasets.
> (3) Baseline Behavior.
> SrvfRegNet+TSLANet reaches competitive accuracy (94.02\%) but fails to produce meaningful alignments (ATV $=12.8$). DeepFRC uniquely delivers strong classification and high-quality registration simultaneously, even at scale.
> These new results, now included in Section 5, demonstrate that DeepFRC maintains its advantages in both performance and interpretability as the dataset size grows.

---

### Author Response · Authors · 2025-12-01
**Final Author Summary**

We thank all reviewers for their thoughtful and constructive feedback. The manuscript has been substantially improved in response to comments from Reviewers q1ar, gJ51, UBjh, and RLqL, with all revisions highlighted in red in the updated submission.

1. Revisions Made in Response to Reviewers

Across the four reviews, we have incorporated extensive clarifications, new analyses, expanded theoretical discussion, and additional experiments.

(a) New Experiments & Analyses: (1) Added a large-scale robustness study evaluating non-diffeomorphic warpings, showing graceful degradation (Section 5). (2) Expanded computational complexity analysis: training time, FLOPs, and comparisons with TTN and SrvfRegNet-based hybrids (Section 5). (3) Added paired t-tests demonstrating statistical significance of all ablation results (Section 4.2, Table 3). (4) Added a new large-scale benchmark (100× Symbol dataset) assessing performance and scalability across all baselines (Table A4).

(b) New Discussions & Limitations: (1) Added explicit discussion of when joint modeling may be unnecessary, when theoretical assumptions may fail, and how DeepFRC behaves in these settings (Section 6). (2) Added domain-specific interpretation examples for MotionSense and Symbol datasets.

(c) Clarifications & Exposition: (1) Clarified distinctions between DeepFRC and TTN (Section 1.2). (2) Substantially expanded explanations of the diffeomorphic neural deformation operator, monotonicity guarantees, and theoretical assumptions (Sections 2.1 and 3).

These revisions strengthen the paper’s clarity, empirical rigor, and theoretical grounding.

2. Addressing Misunderstandings in Reviewer q1ar’s Report

While Reviewer q1ar offered helpful suggestions, several concerns stemmed from factual misunderstandings:

(a) Misclassification of Zhang & Telesca (2014): The reviewer equated the unsupervised joint clustering–registration model with our supervised joint registration–classification; these are fundamentally distinct.

(b) Baseline Misinterpretation: Contrary to the claim that recent baselines were missing, the original manuscript already includes TSLANet (2024)—a state-of-the-art architecture—and all known deep functional registration models.

(c) Theoretical Assumptions: The reviewer stated that Lipschitz and compactness conditions were not discussed, but these appear in Section 2.2 and the proof of Theorem 3.3; the revision further clarifies this.

(d) Efficiency Concerns: “Runtime’’ was misinterpreted as inference time; it referred to training time. This has now been corrected and expanded.

We also note that Reviewer gJ51 explicitly stated willingness to raise their score if certain clarification points were addressed. These points—including TTN comparisons, handling non-diffeomorphic warpings, and theoretical assumptions—have now been fully resolved through new experiments and expanded discussion.

3. Contribution and Novelty of the Manuscript

DeepFRC is the first end-to-end deep learning framework for joint diffeomorphic functional registration and supervised classification. Its novelty arises from an integrated architectural design, a new loss formulation, and new theoretical guarantees.

Key contributions:
(a) A unified architecture combining a diffeomorphic neural deformation operator, a smooth spectral representation, and a classifier tightly coupled to the alignment process.
(b) A new class-aware contrastive alignment loss that simultaneously minimizes intra-class SRVF variance and maximizes inter-class separation—unlike existing registration, classification, or contrastive objectives.
(c) The first theoretical generalization guarantees for joint registration–classification (Theorems 3.1 and 3.3), linking registration error to downstream discrimination.
(d) Strong empirical performance: superior alignment quality, state-of-the-art classification accuracy (including over TSLANet-2024), statistically significant ablations, robustness to non-diffeomorphic distortions, and scalability to large datasets.
(e) Interpretability and domain relevance, demonstrated through canonical biomechanical and handwriting alignments.

In summary, this work advances Functional Data Analysis (FDA) by introducing DeepFRC, the first principled, end-to-end framework for joint diffeomorphic registration and supervised classification. The manuscript has been thoroughly revised to incorporate all reviewer feedback, and we believe it now compellingly demonstrates the significance and novelty of DeepFRC as a meaningful advance for the FDA community.

---

### Meta-Review · Area_Chair_Vsip · 2025-12-17

**Summary:**

Reviewers generally agreed that the problem is well-motivated and that the joint registration–classification formulation is interesting, with solid theory and strong empirical results, but raised some concerns about perceived novelty and positioning. Some reviewers viewed the contributions as primarily integrative, questioned whether the baseline set was sufficiently comprehensive and up to date, and asked for clearer justification of efficiency claims and practical downstream impact.

Metareviewer has reviewed the paper, and thinks it has merits to the ICLR community. Given the overall scores are leaning towards acceptance, the metareviewer recommends acceptance (borderline) of this paper.

**Reviewer Concerns:**

The rebuttal and revision addressed most concrete concerns: they clarified why the deformation operator’s monotone cumulative-sum construction yields diffeomorphic warpings, expanded discussion of theoretical assumptions and added robustness tests when diffeomorphism is violated, strengthened positioning vs. TTN and clarified prior-work comparisons, and substantially improved the efficiency evidence (training time/params/FLOPs) plus added statistical significance and clearer dataset/architecture descriptions.

Outstanding concerns are mainly higher-level and remain partly subjective: some reviewers still view the contribution as largely an integration of standard modules, some are unconvinced the baseline set is fully comprehensive, and one reviewer still wanted a clearer “real-world downstream utility” case study beyond the registration+classification benchmarks.

**Reviewer Scores:**

Most reviewer concerns were addressed in the rebuttal and revision: the paper clarified how the cumulative-sum construction enforces monotone (thus invertible) warpings and strengthened the exposition around diffeomorphic guarantees; added missing compute evidence (training time/params/FLOPs and clearer complexity discussion); expanded TTN positioning and provided additional robustness results when true warpings violate diffeomorphism; and added statistical significance tests plus clearer dataset/assumption/limitations discussions.

The main outstanding concerns are higher-level: some reviewers still view the novelty as primarily integration of standard components (despite the new class-aware loss and theory), remain unconvinced that the baseline set is maximally comprehensive, and one reviewer still wanted a more compelling downstream application case study beyond improved registration+classification benchmarks.

Reviewer gJ51 indicated willingness to raise the score if the discussion points were addressed; given the added TTN comparison, robustness-to-misspecification experiment, and explicit limitations (6). Reviewer q1ar’s technical objections (diffeomorphism explanation, efficiency reporting, assumptions/baselines) were directly addressed with clarifications and added compute analysis, so their score could move from borderline reject to borderline accept (4/5), though this depends on whether they accept the “no newer joint FDA baselines” argument. Reviewers PLqL and UBjh were already slightly positive (6) and, since their main questions were answered (assumptions/scaling/efficiency/statistics), they would likely stay at 6 (or at most tick up slightly), especially as UBjh explicitly maintained their score.

---

### Decision · Program_Chairs · 2026-01-26

Accept (Poster)